# NIAQUE: Neural Interpretable Any-Quantile Estimation — Towards Large Probabilistic Regression Models

## Abstract

State-of-the-art computer vision and language models largely owe their success to the ability to represent massive prior knowledge contained in multiple datasets by learning over multiple tasks. However, large-scale cross-dataset studies of deep probabilistic regression models are missing, presenting a significant research gap. To bridge this gap, in this paper we propose, analyze, and evaluate a novel probabilistic regression model, capable of solving multiple regression tasks represented by different datasets. To demonstrate the feasibility of such operation and the efficacy of our model, we define a novel multi-dataset probabilistic regression benchmark LPRM-101. Our results on this benchmark imply that the proposed model is capable of solving a probabilistic regression problem jointly over multiple datasets. The model, which we call NIAQUE, learns a meaningful cross-dataset representation, scores favorably against strong tree-based baselines and Transformer and exhibits positive transfer on unseen datasets after fine-tuning.

## 1 Introduction

For decades, the ML community has focused on addressing tabular predictive modeling problems using advanced, non-linear models. Tree-based methods such as Random Forests (Breiman, 2001), XGBoost (Chen and Guestrin, 2016), LightGBM (Ke et al., 2017), and CatBoost (Prokhorenkova et al., 2019) have traditionally been the preferred approaches for solving these tasks. The first notable shift toward deep learning in large-scale dense tabular problems occurred in domains like e-commerce, ads, and click-through rate modeling, where deep representation learning demonstrated clear advantages (Guo et al., 2017), and TabNet (Arik and Pfister, 2021) emerged as the first deep model built specifically for tabular data. Recent findings based on Transformer architectures highlight that deep learning models typically require extensive upstream pre-training data to perform effectively (Arik and Pfister, 2021; Levin et al., 2023; Hollmann et al., 2023). Our study complements existing results by showing that deep learning models can be trained directly on a large collection of diverse downstream datasets to effectively solve the multi-task learning problem. We show that meaningful dataset-level representations emerge in this setting, and when compared to tree-based approaches under similar conditions, deep probabilistic models clearly outperform them. Additionally, we show that our model pretrained on a large set of datasets exhibits positive transfer on a set of unseen datasets after fine-tuning. These new results establish the viability of cross-dataset multi-task learning and transfer learning, with direct implications for model architecture design in large enterprises. Currently, the common approach involves deploying isolated, disjoint models, each requiring substantial scientific and engineering support. Our findings indicate that unified models capable of concurrently addressing multiple probabilistic regression tasks represent a viable alternative. On top of this, we show the feasibility of pretraining a probabilistic regression model that can then be fine-tuned on the target problem of interest demonstrating positive transfer compared to the model trained on the same target dataset from scratch.

The growing recognition of the importance of probabilistic and distributional modeling in predictive scenarios is evident too, particularly in fields like medical applications, such as clinical trial analysis (Heller et al., 2022). Moreover, representation of uncertainty is a general requirement for any problem with incomplete knowledge (Taylor et al., 1994), and predictive distributions build an understanding of uncertainty. Hence, distributional modeling is a natural choice for overcoming

barriers to ML adoption and enhancing system trustworthiness. A model that can flag its potential failure cases is more trustworthy than the model that is randomly and unpredictably wrong. By quantifying output distributions, probabilistic models can alert downstream users to high-uncertainty cases (e.g., large posterior distribution spreads), where predictions should not be trusted in critical decisions. Another dimension of trust, interpretability, is gaining importance for predictive models in tabular data (Sahakyan et al., 2021). In this paper, we focus on global interpretability—identifying independent variables that are key to solving a given problem. We show that probabilistic modeling and feature importance assessment can work in tandem: the posterior distribution of individual features helps highlight those that strongly impact prediction accuracy.

In this work, we identify and bridge several key research gaps. First, existing multi-dataset tabular benchmarks are predominantly focused on classification problems, lacking a comprehensive benchmark for large-scale probabilistic regression tasks. We introduce a new multi-dataset regression benchmark and train multiple baseline models across all its datasets in a multi-task fashion. This benchmark comprises 101 diverse datasets from various domains, with varying sample sizes and feature dimensions. Second, we propose NIAQUE, a novel probabilistic regression model capable of solving multi-task learning problem across multiple diverse datasets, effectively developing meaningful dataset-level representations. NIAQUE compares favorably against strong tree-based baselines and Transformers, despite being trained solely on a collection of downstream regression tasks. Moreover, it demonstrates positive transfer when pretrained on a large collection of regression datasets and later fine-tuned on unseen new datasets. Our contributions can be summarized as follows.

- We define a new probabilistic regression benchmark based on 101 diverse regression datasets publicly available from UCI, PMLB, OpenML and Kaggle repositories

- We introduce NIAQUE, a novel model designed to address probabilistic regression by learning to approximate the inverse of the posterior distribution during training.

- Our theoretical analysis provides strong methodological foundation for NIAQUE.

- We demonstrate that NIAQUE achieves superior accuracy compared to strong baselines

- We propose feature weights derived from NIAQUE's marginal posterior distributions that enhance interpretability by taking advantage of the model's probabilistic nature.

## 1.1 RELATED WORK

**Multi-task learning** has been modus operandi in computer vision (Sun et al., 2021; Radford et al., 2021) and language modeling (Devlin et al., 2019). More recently, cross-dataset learning has been applied to univariate time-series forecasting (Garza and Mergenthaler-Canseco, 2023; Ansari et al., 2024). In the context of tabular data processing, the emphasis so far has been on classification problems and point (non-distributional) regressions. For example, Transformer is compared with tree-based models on a collection of 20 and 67 classification datasets, respectively, in a series of papers (Müller et al., 2022; Hollmann et al., 2023), MLP is compared against Tabnet and trees on 40 classification datasets in (Kadra et al., 2021). Similarly, (Grinsztajn et al., 2022) compares Transformer and a few other architectures (ResNet, MLP) against tree-based models on 45 dataset benchmark. It is important to note that only about half of the 45 datasets are regression datasets and models are fitted to each dataset independently. While (Hollmann et al., 2023) and (Grinsztajn et al., 2022) agree that Transformer is the strongest model for tabular data among deep learning models, the latter concludes tree-based models to be the ultimate winners on performance while the former present evidence in favor of Transformers. Finally, Salinas and Erickson (2023) present a large tabular benchmark, but only 28 of the datasets represent regression problems.

In terms of **neural modeling methodology**, our work is closely related to (Oreshkin et al., 2022), who used a similar architecture in the context of human pose completion in animation. We extend this architecture with the any-quantile modeling and show interesting theoretical properties of the proposed approach. Other permutation invariant architectures for encoding unstructured variable inputs are also related. Attention models (Bahdanau et al., 2015) and Transformer (Vaswani et al., 2017) have been proposed in the context of natural language processing. Prototypical networks (Snell et al., 2017) use average pooled embedding to encode semantic classes in few-shot image classification. PointNet (Qi et al., 2017) and DeepSets (Zaheer et al., 2017) represent variable input dimension by max-pooling MLP output in the context of 3D point clouds and text concept retrieval, further

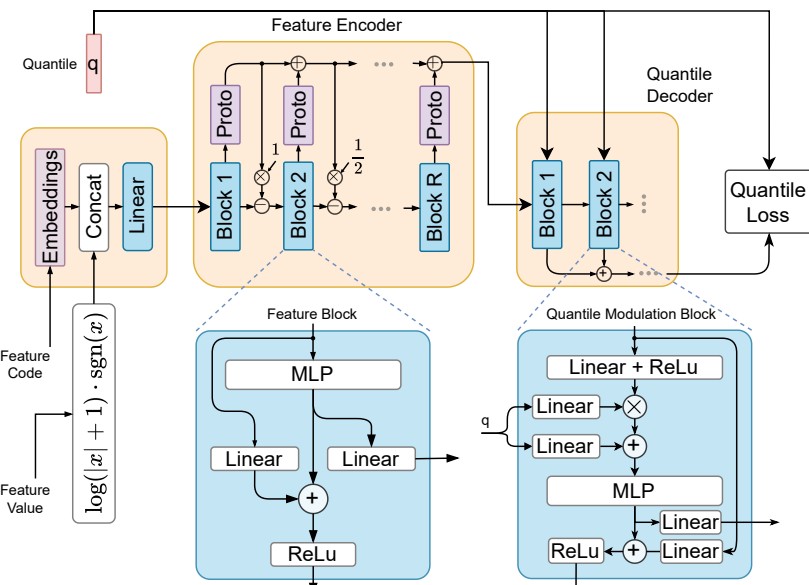

Figure 1: NIAQUE architecture accepts variable-dimension independent variable transforming it to the fixed-size representation, thus enabling its operation across diverse multi-task regression datasets.

generalized by Niemeyer et al. (2019) resulting in ResPointNet architecture. From a probabilistic modeling perspective, this work builds on the electricity forecasting framework proposed by Smyl et al. (2024), advancing both the theoretical foundations and neural modeling techniques. Our contributions extend the applicability of these methods to general cross-dataset conditional probabilistic regression problems. Alternative approaches, such as Neural Processes (Garnelo et al., 2018b) and Conditional Neural Processes (Garnelo et al., 2018a), also generate conditional probabilistic solutions to regression problems. However, these methods are limited to fixed-dimensional input spaces and are not directly applicable to the cross-dataset, multi-task learning problem addressed here, where datasets vary in the number of independent variables. Moreover, unlike Garnelo et al. (2018b) and Garnelo et al. (2018a), our approach demonstrates the ability to transfer knowledge to entirely new datasets, even when their dependent variable domains do not overlap with the training data.

## 1.2 PROBABILISTIC REGRESSION PROBLEM

We consider the problem of estimating the underlying dependent variable $y \in \mathbb{R}$ given a variable set of independent variables captured in vector $\mathbf{x}$ of variable dimensionality. The relationship between dependent and independent variables is assumed to be captured by an unknown non-linear function $\Psi$ and stochastic noise $\varepsilon$ with unknown distribution:

$$y = \Psi(\mathbf{x}, \varepsilon) \tag{1}$$

The formulation of regression problem provided above is very general and this motivates us to also define its solution in a general non-parameteric form. In particular, we further define the probabilistic regression solution using a non-linear regression function $f_\theta : \mathbb{R}^{|\mathbf{x}| \times Q} \to \mathbb{R}^Q$, parameterized with $\theta \in \Theta$, predicting a $Q$-tuple of $q$-th quantiles of the unknown dependent variable based on available observation $\mathbf{x}$. The accuracy of distributional dependent variable prediction is evaluated using Continuous Ranked Probability Score:

$$\mathrm{CRPS}(F, y) = \int_{\mathbb{R}} \left(F(z) - \mathbb{1}_{\{z \geq y\}}\right)^2 \mathrm{d}z, \tag{2}$$

where $y$ is the dependent variable value and $F$ denotes the cumulative distribution function (CDF) derived from the predicted set of quantiles, $\mathbb{1}$ denotes the indicator function.

## 2  NIAQUE

In this section we first outline the proposed general solution to the probabilistic regression problem based on training a machine learning model using any-quantile approach. We further provide the theoretical analysis showing that the training using proposed methodology has inverse cumulative distribution function of the data as the optimal solution.

### 2.1  ANY-QUANTILE LEARNING

The any-quantile learning methodology depicted in Figure 1 asserts that both model and the loss function shall accept quantile level $q$ as input, making the model $q$-programmable. Therefore, at inference time the user of the model has the flexibility of querying the model with any combination of target quantiles that best suit the user's downstream application. Let $y$ represent the observed value, $\widehat{y}_q$ the predicted $q$-quantile, and suppose the model is trained using quantile loss:

$$\rho(y, \widehat{y}_q) = \begin{cases} (y - \widehat{y}_q)q & \text{if } y \geq \widehat{y}_q \\ (y - \widehat{y}_q)(q - 1) & \text{otherwise} \end{cases}. \tag{3}$$

We consider that the model is trained on $S$-sample dataset of $(\mathbf{x}, y)$ tuples derived from the joint distribution $P_{y,\mathbf{x}}$. We also assume, without loss of generality, that training is conducted using stochastic gradient descent (SGD) with a mini-batch size of $B$, and that the quantile value $q$ is sampled from $U(0, 1)$. This results in the following model parameter update at iteration $k$:

$$\theta_{k+1} = \theta_k - \eta_k \nabla_\theta \frac{1}{B} \sum_{i=1}^{B} \rho(y_i, f_\theta(\mathbf{x}_i, q_i)). \tag{4}$$

Sequence $\theta_k$ converges to the optimum over the full training dataset of size $S$ Karimi et al. (2016):

$$\theta^* = \arg\min_{\theta \in \Theta} \frac{1}{S} \sum_{i=1}^{S} \rho(y_i, f_\theta(\mathbf{x}_i, q_i)). \tag{5}$$

By the strong law of large numbers, as $S$ increases without bound, the sum in the last equation converges to the following w.p. 1:

$$\mathbb{E}_{\mathbf{x},y} \mathbb{E}_q \rho(y, f_\theta(\mathbf{x}, q)) = \mathbb{E}_{\mathbf{x},y} \int_0^1 \rho(y, f_\theta(\mathbf{x}, q)) dq. \tag{6}$$

Lastly, we note that besides the L2 formulation (2), CRPS can also be expressed in its integral form using the inverse CDF $F^{-1}$ (Gneiting and Ranjan, 2011):

$$\text{CRPS}(F, y) = 2 \int_0^1 \rho(y, F^{-1}(q)) dq. \tag{7}$$

Based on this fact, the following theorem proves that the expected pinball loss (6) is minimized when $f_\theta(\mathbf{x}, q)$ corresponds to the inverse of the posterior CDF $P_{y|\mathbf{x}}$.

**Theorem 1.** *Let $F$ be a probability measure over variable $y$ such that inverse $F^{-1}$ exists and let $P_{y,\mathbf{x}}$ be the joint probability measure of variables $\mathbf{x}, y$. Then the expected loss, $\mathbb{E}_{\mathbf{x},y,q}\, \rho(y, F^{-1}(q))$, is minimized if and only if $F = P_{y|\mathbf{x}}$.*

*Proof.* The proof is in Appendix A. □

This leads to the following conclusions. First, the SGD update based on quantile loss (4) optimizes the empirical risk (5) corresponding to the expected loss (6). Based on (6,7) and Theorem 1, $f_{\theta^\star} = \arg\min_{f_\theta} \mathbb{E}_{\mathbf{x},y,q}\, \rho(y, f_\theta(\mathbf{x}, q))$, has a clear interpretation as the inverse CDF corresponding to $P_{y|\mathbf{x}}$. Second, as $k$ (the SGD iteration index) and $S$ (training sample size) increase, and if in addition $f_\theta$ is implemented as an MLP whose width and depth scale appropriately with sample size $S$, then (Farrell et al., 2021, Theorem 1) implies that the SGD solution also converges to $f_{\theta^\star}(\mathbf{x}, q) \equiv P_{y|\mathbf{x}}^{-1}(q)$. In other words, given uniform sample $q \sim U(0, 1)$, $\widehat{y}_q = f_{\theta^\star}(\mathbf{x}, q)$ has the interpretation of the sample from the posterior distribution of $y$, $\widehat{y} \sim p(y|\mathbf{x})$, which obviously follows from the proof of the inversion method (Devroye, 1986, Theorem 2.1).

## 2.2 NEURAL ARCHITECTURE

NIAQUE, shown in Fig. 1, follows the encoder-decoder pattern. Encoder deals with $N$ independent variables, where $N$ is variable. At inference time, for $i$-th observation sample, $\mathbf{x}_i$, with variable dimensionality $N_i$ it accepts a tensor of values of dimensionality $1 \times N_i$ and a tensor of feature codes of dimensionality $1 \times N_i$, transforms, embeds and concatenates them into tensor of size $1 \times N_i \times E_{in}$. The encoder then collapses the independent variable dimension using prototype approach, resulting in output embedding of size $1 \times E$. Decoder modulates the quantile agnostic representation received from encoder with the vector of quantiles $\mathbf{q} \in \mathbb{R}^Q$, again, of arbitrary dimensionality $Q$. This design is compute efficient with complexity $O(N_i + Q)$ for a given $\mathbf{x}_i$, whereas processing quantiles and observations in encoder and decoder would imply complexity $O(N_i Q)$.

**Inputs.** For each element in the observation vector $\mathbf{x}$, NIAQUE receives its value along with an integer representing the independent variable ID, wrapped with a learnable embedding. The variable ID is crucial for capturing the distinct statistical properties of each variable, the interactions between independent variables, and their statistical relationship with the dependent variable. The embedded variable ID is concatenated with its value, transformed into the log domain:

$$z = \log(|x| + 1) \cdot \operatorname{sgn}(x) \tag{8}$$

Log-transform aligns the dynamic range of variable value with that of ID embeddings and preserves the sign, which is important to make training successful (this intuition is confirmed by ablation).

**Observation Encoder** is structured as a two-loop residual network. We first present the encoder equations, followed by a detailed explanation of the underlying architectural motivations, dropping sample index $i$ for brevity. We assume the encoder input to be $\mathbf{x}_1 = \mathbf{x}_{in} \in \mathbb{R}^{N \times E_{in}}$, where $E_{in}$ is the size of embedding vector for each independent variable, omitting the batch dimension for brevity. In this case, the fully-connected layer $\operatorname{FC}_{r,\ell}$, with $\ell = 1...L$, in the residual block $r$, $r = 1 \ldots R$, with weights $\mathbf{W}_{r,\ell}$ and biases $\mathbf{a}_{r,\ell}$ can be conveniently described as $\operatorname{FC}_{r,\ell}(\mathbf{h}_{r,\ell-1}) \equiv \operatorname{RELU}(\mathbf{W}_{r,\ell}\mathbf{h}_{r,\ell-1} + \mathbf{a}_{r,\ell})$. Given prototype layer definition, $\operatorname{PROTOTYPE}(\mathbf{x}) \equiv \frac{1}{N}\sum_{i=1}^{N}\mathbf{x}[i,:]$, the observation encoder can be described as:

$$\mathbf{x}_r = \operatorname{RELU}(\mathbf{b}_{r-1} - 1/(r-1) \cdot \mathbf{p}_{r-1}), \tag{9}$$

$$\mathbf{h}_{r,1} = \operatorname{FC}_{r,1}(\mathbf{x}_r), \ \ldots, \ \mathbf{h}_{r,L} = \operatorname{FC}_{r,L}(\mathbf{h}_{r,L-1}), \tag{10}$$

$$\mathbf{b}_r = \operatorname{RELU}(\mathbf{L}_r\mathbf{x}_r + \mathbf{h}_{r,L}), \ \mathbf{f}_r = \mathbf{F}_r\mathbf{h}_{r,L}, \tag{11}$$

$$\mathbf{p}_r = \mathbf{p}_{r-1} + \operatorname{PROTOTYPE}(\mathbf{f}_r). \tag{12}$$

Equations (10) and (11) implement the MLP and the first residual loop. The second residual mechanism, described in equations (9) and (12), is motivated by the following. First, equation (12) aggregates the forward encoding of individual independent variables into a prototype-based representation of the overall observation vector. Second, equation (9) enforces an inductive bias, ensuring that information from independent variables is only significant when it deviates from the existing observation embedding, $\mathbf{p}_{r-1}$, by applying a delta-mode constraint. Finally, the representation of observations is accumulated across residual blocks in (12), effectively implementing skip connections.

**Quantile Decoder** is the fully-connected conditioned residual architecture depicted in Fig. 1 (top right) consisting of the conditioned MLP blocks appearing in Fig. 1 (bottom right). The quantile value is injected inside the MLP block using FiLM modulation principle (Perez et al., 2018). Quantile Decoder takes the observation embedding, $\widetilde{\mathbf{b}}0 = \mathbf{p}R \in \mathbb{R}^E$, and generates quantile-modulated representations, $\widetilde{\mathbf{f}}_R \in \mathbb{R}^{Q \times E}$, for all quantiles $\mathbf{q} \in \mathbb{R}^Q$, using the following set of equations:

$$\mathbf{h}_{r,1} = \operatorname{FC}_{r,1}^{\operatorname{QD}}(\widetilde{\mathbf{b}}_{r-1}), \quad \gamma_r, \beta_r = \operatorname{LINEAR}_r(\mathbf{q})$$

$$\mathbf{h}_{r,2} = \operatorname{FC}_{r,1}^{\operatorname{QD}}((1 + \gamma_r) \cdot \mathbf{h}_{r,1} + \beta_r), \ \ldots, \ \mathbf{h}_{r,L} = \operatorname{FC}_{r,L}^{\operatorname{QD}}(\mathbf{h}_{r,L-1}), \tag{13}$$

$$\widetilde{\mathbf{b}}_r = \operatorname{RELU}(\mathbf{L}_r^{\operatorname{QD}}\widetilde{\mathbf{b}}_{r-1} + \mathbf{h}_{r,L}), \ \widetilde{\mathbf{f}}_r = \widetilde{\mathbf{f}}_{r-1} + \mathbf{F}_r^{\operatorname{QD}}\mathbf{h}_{r,L}.$$

The final prediction, $\widehat{\mathbf{y}}_q \in \mathbb{R}^Q$, is generated via linear projection, $\widehat{\mathbf{y}}_q = \operatorname{LINEAR}[\widetilde{\mathbf{f}}_r]$.

## 2.3 INTERPRETABILITY

The core feature of NIAQUE is its probabilistic formulation, which enables prediction of any quantile of the dependent variable conditioned on any combination of available independent variables.

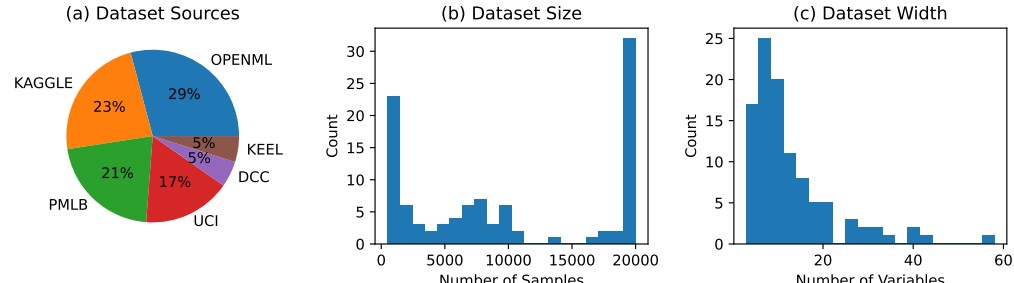

Figure 2: Summary statistics of the LPRM-101 benchmark. (a) The distribution by dataset sources, (b) the distribution of dataset sizes, (c) the distribution of variable count per dataset.

Consider $f_\theta(\mathbf{x}_s, q)$ to be NIAQUE prediction of quantile $q$ when only independent variable $\mathbf{x}_s$ is provided. We can then define the posterior confidence interval $\mathrm{CI}_{\alpha,s} = f_\theta(\mathbf{x}_s, 1-\alpha/2) - f_\theta(\mathbf{x}_s, \alpha/2)$. Confidence interval defines the width of the region in which the ground truth will fall with probability $1 - \alpha$. Independent variables that are stronger predictors will tend to produce narrower confidence intervals. Therefore, we should be able to identify globally important variables by calculating the average width of their confidence intervals and comparing it against that of other variables. Based on this simple intuition, for the independent variable $s$, we define the normalized weight $W_s$ as:

$$W_s = \frac{\overline{W}_s}{\sum_s \overline{W}_s}, \quad \overline{W}_s = \frac{1}{\overline{\mathrm{CI}}_{0.95,s}}; \tag{14}$$

$$\overline{\mathrm{CI}}_{\alpha,s} = \frac{1}{S} \sum_i f_\theta(y_{s,i}, 1 - \alpha/2) - f_\theta(y_{s,i}, \alpha/2). \tag{15}$$

Note that $\overline{\mathrm{CI}}_{\alpha,s}$ is the average width of posterior confidence interval over datapoints $y_{s,i}$. We propose to use a validation dataset for computing this quantity. The proposed feature weight depends on the accuracy of marginal distribution modeling. To better model the marginal distributions of individual features and enable the proposed interpretability mechanism, we augment the dataset by adding rows that contain only a single feature, constituting approximately 5% of the total training data. Our ablation study shows that this is an important step enabling the proposed interpretability mechanism.

### 2.4 NIAQUE as a Cross-Dataset Multi-Task Probabilistic Regression Model

NIAQUE handles variable input combinations through the use of semantically encoded variables. Thus, the model can be trained across multiple heterogeneous datasets by presenting to the model, for each dataset, only the relevant variables. The combination of the variables informs the model of the dataset and task required for inference via their learnable semantic embeddings. When pretrained across multiple datasets, the model is expected to generalize effectively to each dataset. Additionally, the pretrained model is expected to generalize to new unseen datasets with appropriate additional fine-tuning. Our experimental results provide empirical validation to both these hypotheses.

## 3 LPRM-101 Benchmark

LPRM-101 is the multi-dataset benchmark for large probabilistic regression models (hence, LPRM) consisting of 101 dataset (hence LPRM-101). The datasets, along with their sample count, number of variables and source information are listed in Table 3 of Appendix B. To construct the benchmark, we first collect 101 dataset publicly available from the following primary repositories: UCI (Kelly et al., 2017), Kaggle (Kaggle, 2024), PMLB (Romano et al., 2021; Olson et al., 2017), OpenML (Vanschoren et al., 2013), KEEL (Alcalá-Fdez et al., 2011). We focus specifically on the regression task in which the dependent variable is continuous or, if it has limited number of levels, these are ordered such as student exam scores or wine quality. The target variable in each dataset is normalized to the [0, 10] range and the independent variables are used as is, raw. The target variable scaling is applied to equalize the contributions of samples from each dataset to the evaluation metrics. Datasets have variable number of samples, the lowest being just below 1000. For very large datasets we limit the number of samples used in our benchmark to be 20,000 by subsampling uniformly at random. This allows us (i) to model task imbalance, and at the same time (ii) avoid the situation in which a few

large datasets could completely dominate the training and evaluation of the model. The distribution of datasets by source, number of samples and number of variables is shown in Figure 2.

For evaluating the prediction accuracy we use the following point prediction accuracy metrics: sMAPE, AAD, RMSE, BIAS and distributional prediction accuracy metrics: CRPS and COVERAGE. We implement the 0.8/0.1/0.1 training/validation/test split sampled uniformly at random using stratified sampling at the level of each dataset. This approach mitigates the risk of disproportionately including a large number of samples from a larger dataset in the validation/test splits, while potentially excluding samples from smaller datasets due to sampling chance. Evaluation metrics are averaged over all samples in the test split containing samples from all datasets. The ground truth sample is denoted as $y_i$ and it's $q$-th quantile prediction as $\hat{y}_{i,q}$. Given the $N$-sample dataset, the point prediction accuracy metrics are defined as:

$$\text{sMAPE} = \frac{200}{S} \sum_{i=1}^{S} \frac{|y_i - \hat{y}_{i,0.5}|}{|y_i| + |\hat{y}_{i,0.5}|}, \quad \text{AAD} = \frac{1}{S} \sum_{i=1}^{S} |y_i - \hat{y}_{i,0.5}|, \tag{16}$$

$$\text{RMSE} = \sqrt{\frac{1}{S} \sum_{i=1}^{S} (y_i - \hat{y}_{i,0.5})^2}, \quad \text{BIAS} = \frac{1}{S} \sum_{i=1}^{S} \hat{y}_{i,0.5} - y_i \tag{17}$$

The distributional accuracy metrics are defined over a random set of $Q = 200$ quantiles sampled uniformly at random and are formally defined as follows:

$$\text{CRPS} = \frac{2}{SQ} \sum_{i=1}^{S} \sum_{j=1}^{Q} \rho(y_i, \hat{y}_{i,q_j}), \tag{18}$$

$$\text{COVERAGE} @ \alpha = \frac{100}{S} \sum_{i=1}^{S} \mathbb{1}[y_i > \hat{y}_{i,0.5-\alpha/200}] \mathbb{1}[y_i < \hat{y}_{i,0.5+\alpha/200}]. \tag{19}$$

## 4  EMPIRICAL RESULTS

Our empirical results are obtained on the LPRM-101 benchmark introduced in Section 3. The key quantitative result appears in Table 1. We compare NIAQUE against a number of tree-based baselines XGBoost (Chen and Guestrin, 2016), LightGBM (Ke et al., 2017), CatBoost (Prokhorenkova et al., 2019). XGBoost and CatBoost are trained on the multi-quantile loss with fixed quantiles (additional quantiles required for evaluation are linearly interpolated). LightGBM does not support multi-quantile loss and we trained one model per quantile (similarly, XGBoost trains one model per quantile under the hood). Models trained on all 101 datasets are denoted by the suffix *global*, while those with the suffix *local* are trained individually on each dataset. Transformer baseline and ablations are discussed in detail in Appendix G, including the architectural diagram. The gist of it is that the original Transformer's encoder/decoder structure (Vaswani et al., 2017) replaces NIAQUE's feature encoder, while the quantile decoder and training procedure are kept to be exactly the same as those of NIAQUE.

**Training Details** All global models are trained by drawing cases from the train splits of all datasets jointly and uniformly at random. To train tree-based global models, we joined all datasets resulting in a large flat table, whose rows contain samples from all datasets and whose columns contain features from all datasets. The row-column locations corresponding to features that do not exist in a given dataset are filled with NA values. NIAQUE and Transformer are trained using the loss in eq. (3) and Adam optimizer with initial learning rate 0.0001 that steps down by a factor of 10 at 500k, 600k and 700k batches, training for total 500 epochs. In a batch of 512 instances, a quantile, $q$, is generated uniformly at random for each instance. For both Transformer and NIAQUE models we found that feature dropout with rate 0.2 implemented as discussed in more detail in Appendix I helped to improve accuracy. Training NIAQUE and Transformer models on 4xV100 GPUs requires approximately 24 and 48 hours, respectively. XGBoost training time on 1xV100 is about 30min on 3 quantiles and grows linearly with the number of quantiles.

**Multi-Task Learning Experiment** results are reported in Table 1. Detailed ablation studies of all models are reported in Tables 5-10 of Appendices E-I. The results suggest a negative correlation

Table 1: Accuracy of the proposed NIAQUE approach compared to the tree-based baselines and Transformer on LPRM-101 benchmark. Smaller values for SMAPE, AAD, RMSE, CRPS are better. BIAS values closer to zero are better. COVERAGE @ 95 values closer to 95 are better. The results with confidence intervals derived from 4 random seed runs are presented in Appendix C, Tables 4,5.

| | SMAPE | AAD | BIAS | RMSE | CRPS | COVERAGE @ 95 |
|---|---|---|---|---|---|---|
| XGBoost-global | 31.4 | 0.574 | -0.15 | 1.056 | 0.636 | 94.6 |
| XGBoost-local | 25.6 | 0.433 | -0.03 | 0.883 | 0.334 | 90.8 |
| LightGBM-global | 27.5 | 0.475 | -0.06 | 0.930 | 0.426 | 94.8 |
| LightGBM-local | 25.7 | 0.427 | -0.03 | 0.865 | 0.327 | 91.5 |
| CATBOOST-global | 31.3 | 0.561 | -0.12 | 1.030 | 0.443 | 94.9 |
| CATBOOST-local | 24.3 | 0.408 | -0.03 | 0.840 | 0.315 | 92.7 |
| Transformer-local | 26.9 | 0.462 | -0.05 | 0.904 | 0.329 | 93.6 |
| Transformer-global | 23.1 | 0.383 | -0.01 | 0.806 | 0.272 | 94.6 |
| NIAQUE-local | 22.8 | 0.377 | -0.03 | 0.797 | 0.267 | 94.9 |
| NIAQUE-global | 22.1 | 0.367 | -0.02 | 0.787 | 0.261 | 94.6 |

between the quality of distributional predictions, as measured by the COVERAGE @ 95 metric, and point prediction accuracy metrics (e.g., SMAPE, AAD, RMSE). Therefore, to ensure a fair comparison, Table 1 presents the best result for each model, constrained by a COVERAGE @ 95 value within the [94.5, 95.5] range. For models unable to meet this criterion, the results reflect the case where their COVERAGE @ 95 is closest to 95. Overall, our results demonstrate the following key findings. First, NIAQUE effectively addresses the distributional modeling task while maintaining state-of-the-art point prediction accuracy. Second, tree-based models struggle to achieve both point and distributional accuracy simultaneously. Furthermore, tree-based models perform better on point prediction tasks in the local training setting, but experience a decline in both point accuracy (measured by SMAPE, AAD, RMSE) and distributional accuracy (CRPS) under global training. In contrast, neural models represented by NIAQUE benefit from multi-task training across multiple datasets, showing improvements in both point and distributional predictions, even when the datasets are largely unrelated (cf. NIAQUE-local and NIAQUE-global). The multi-task learning experiment establishes the ability of our model to operate effectively across multiple datasets representing multiple tasks.

**Transfer Learning Experiment** conducted in the current section provides further evidence that the learnings from one set of regression datasets can be transferred on another, unseen set of regression datasets. The setup is the following. We divide the overall LPRM-101 benchmark, uniformly at random, into the set of 80 pretraining datasets and the set of 21 unseen test datasets. The baseline control model (NIAQUE-scratch) is trained on each of the unseen 21 datasets from scratch. The treatment model (NIAQUE-pretrained) is first pretrained on 80 pretraining datasets and then fine-tuned on each of the 21 datasets using 10-times smaller learning rate (a common scenario in transfer learning). To provide for a more comprehensive comparison under transfer learning scenario we evaluate the accuracy of fine-tuned and scratch models by subsampling the training portion of held-out datasets with variable rate $p_s$. As $p_s$ decreases, the unseen fine-tuning dataset size shrinks. The test sets are kept constant for apple-to-apple comparison. Metrics of both models are presented in Table 2. Our results demonstrate that the pre-trained model is always more accurate than the model trained from scratch. Pretraining lift increases as the fine-tuning datasets shrink (corresponging to smaller $p_s$). This demonstrates the value of pretraining probabilistic regression models in multi-task fashion and confirms that the learnings on various probabilistic regression tasks are generalizeable and can be transferred on unseen regression datasets. It is important to note that the metrics reported in Tables 1 and 2 are not directly comparable, as the former evaluates performance on 101 datasets, whereas the latter focuses on 21 held-out datasets.

**Representation Analysis**. Figure 3a depicts UMAP projections (McInnes et al., 2018) of row embeddings of all datasets derived from the output of NIAQUE feature encoder and colored by dataset. Clearly, NIAQUE produces meaningful representations of dataset rows that cluster by dataset.

Table 2: Transfer learning results on LPRM-101 benchmark. 80 datsets are randomly sampled for pretraining. Pretrained model is further fine-tuned on 21 held-out datasets whose test splits are used for evaluation. $p_s$ designates the proportion of samples in held-out training datasets used for fine-tuning. Smaller values for SMAPE, AAD, RMSE, CRPS are better. BIAS values closer to zero are better. COVERAGE @ 95 values closer to 95 are better.

|  | $p_s$ | SMAPE | AAD | BIAS | RMSE | CRPS | COVERAGE @ 95 |
|---|---|---|---|---|---|---|---|
| NIAQUE-scratch | 1.0 | 19.4 | 0.49 | -0.04 | 0.96 | 0.351 | 94.4 |
|  | 0.5 | 20.8 | 0.54 | 0.02 | 1.04 | 0.383 | 93.1 |
|  | 0.25 | 21.7 | 0.56 | -0.04 | 1.06 | 0.392 | 94.4 |
|  | 0.1 | 24.7 | 0.60 | -0.04 | 1.10 | 0.423 | 93.0 |
|  | 0.05 | 28.0 | 0.71 | -0.06 | 1.23 | 0.488 | 93.3 |
| NIAQUE-pretrained | 1.0 | 17.7 | 0.47 | -0.04 | 0.94 | 0.334 | 94.6 |
|  | 0.5 | 18.7 | 0.50 | -0.06 | 0.97 | 0.354 | 93.9 |
|  | 0.25 | 20.3 | 0.54 | -0.06 | 1.04 | 0.380 | 94.2 |
|  | 0.1 | 21.9 | 0.57 | -0.07 | 1.08 | 0.404 | 94.4 |
|  | 0.05 | 23.5 | 0.61 | -0.06 | 1.11 | 0.427 | 95.3 |

We conclude that it is viable to train NIAQUE across datasets, resulting in a shared representation space that is discriminative of the regression tasks encapsulated in each dataset.

**Interpretability Analysis**. Figure 3b depicts the empirical analysis of the feature importance assessment mechanism proposed in Section 2.3. The procedure boils down to computing the normalized inverse average confidence interval on the samples from the marginal distribution of each feature drawn from the validation set. Then features are ordered by the importance weight, per dataset. In Figure 3b, top-1 refers to the feature with highest weight, bot-1 refers to the feature with lowest weight. Top-rated (most important) features contribute the most to the AAD metric decrease, when removed. Unimportant features have much smaller effect on AAD. This shows the efficacy of the proposed feature importance assessment in that it produces scores predictive of the effect of features on accuracy. Note that this mechanism is tightly linked to the probabilistic nature of the model, it can be executed on a pre-trained model and it does not require ground truth labels.

**Ablation Studies**. Detailed architecture and training ablations for NIAQUE are presented in Appendix I, demonstrating the following important observations. First, applying the log-transform to input values, as shown in eq. (8), enhances both training stability and prediction accuracy. Second, NIAQUE's performance shows relatively low sensitivity to network width variations, but is more dependent on the number of blocks. Third, the training approach incorporating single-feature rows, which supports the interpretability mechanism discussed in Section 2.3, proves crucial. When single-feature rows are excluded from the training mix (Appendix I, Figure 7c), the model poorly distinguishes between high-importance and low-importance features. However, including single-feature rows to NIAQUE's training mix, creates a clear accuracy gap between the cases of top-importance feature removal and the bottom-importance feature removal. Importantly, this training procedure adjustment does not negatively impact prediction accuracy.

## 5 DISCUSSION

We believe that our results applying NIAQUE to the multi-dataset benchmark LPRM-101 lay out the stepping stone for the development of probabilistic meta-models eventually possessing the following key properties. **Scalability**: A unified model shares computational resources to address multiple regression tasks, optimizing resource utilization and reducing the operational costs of maintaining separate models. **Data Efficiency**: Training on diverse tasks introduces strong regularization effects, and we expect existing datasets to be repurposed to solve emerging problems, promoting data reuse and recycling. **Representation and Generalization**: A model trained across multiple datasets uncovers generalizable representations of regression tasks and ways of solving them, acquiring the ability to apply this knowledge across datasets.

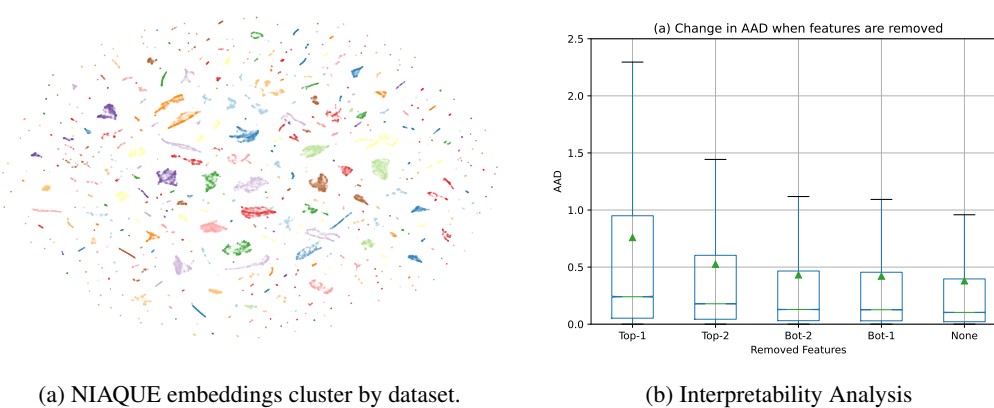

(a) NIAQUE embeddings cluster by dataset.    (b) Interpretability Analysis

Figure 3: **Representation analysis (left)** depicts UMAP projections of embeddings derived from the output of the NIAQUE feature encoder for each sample in LPRM-101 and colored by dataset. Dataset-level clustering of embeddings is evident as points belonging to the same dataset form distinct clusters. **Interpretability Analysis (right)**, NIAQUE accuracy response to the removal of input features according to their importance: top-1 and bot-1 refer to features with the highest and lowest importance scores, respectively. The top-rated features have the greatest impact on AAD degradation when removed, whereas unimportant features exhibit a smaller effect on AAD.

**Limitations**. While we significantly expand the scope of cross-dataset probabilistic model training by applying our neural model to a 101-dataset benchmark, this remains a limited effort. It is still unclear how many datasets are required for a regression model to be considered foundational for solving, for instance, 80% of industry problems. What level of dataset diversity is necessary? Will millions or billions of unrelated datasets be required, or would 10,000 overlapping datasets suffice? Defining and evaluating global success in this context remains an open question, necessitating further research.

**Broader Impacts**. Our findings have implications for designing machine learning deployments based on unified models that address multiple regression tasks. We expect that this will eventually lead to improved operational efficiency and accuracy of the models. However, this could also contribute to the centralization of power among a few large entities. In this context, risk mitigation strategies include (i) improving model computational efficiency and (ii) publicly releasing data, model training code and pretrained models. Additionally, multi-task learning on multiple datasets may introduce new biases not present in locally trained models, making interpretability and fairness research critical. We explore some interpretability aspects in this paper, and further research on interpretability and fairness in large probabilistic regression models pretrained across multiple datasets seems to be an important area for future work.

## 6 CONCLUSIONS

In this paper we introduce NIAQUE, a novel probabilistic regression model, and LPRM-101, a novel multi-dataset large regression model benchmark. We show that learning a probabilistic regression model across datasets is viable and that there exists a strong neural baseline model that compares favorably against usual suspects in the domain of tabular learning: boosted trees and Transformer. We also show that the probabilistic nature of the proposed model opens up a way for achieving global model interpretability via feature importance defined through the average marginal posterior confidence interval. Future work will focus on finding more effective ways of representing variable relationships across datasets, increasing the volume of datasets and applying developed techniques to wide array of application domains, such as multi-variate cross-dataset time series forecasting.

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

## A PROOF OF THEOREM 1

**Theorem.** *Let $F$ be a probability measure over variable $y$ such that inverse $F^{-1}$ exists and let $P_{y,\mathbf{x}}$ be the joint probability measure of variables $\mathbf{x}, y$. Then the expected loss, $\mathbb{E}\,\rho(y, F^{-1}(q))$, is minimized if and only if:*

$$F = P_{y|\mathbf{x}}.  \tag{20}$$

*Additionally:*

$$\min_F \mathbb{E}\,\rho(y, F^{-1}(q)) = \mathbb{E}_{\mathbf{x}} \frac{1}{2} \int_{\mathbb{R}} P_{y|\mathbf{x}}(z)(1 - P_{y|\mathbf{x}}(z))dz.  \tag{21}$$

*Proof.* First, combining (6,7) with the L2 representation of CRPS (2) we can write:

$$\mathbb{E}\,\rho(y, F^{-1}(q)) = \mathbb{E}_{\mathbf{x},y} \frac{1}{2} \int_{\mathbb{R}} \left(F(z) - \mathbb{1}_{\{z \geq y\}}\right)^2 dz  \tag{22}$$

$$= \mathbb{E}_{\mathbf{x}} \mathbb{E}_{y|\mathbf{x}} \frac{1}{2} \int_{\mathbb{R}} F^2(z) - 2F(z)\mathbb{1}_{\{z \geq y\}} + \mathbb{1}_{\{z \geq y\}} dz  \tag{23}$$

$$= \mathbb{E}_{\mathbf{x}} \frac{1}{2} \int_{\mathbb{R}} F^2(z) - 2F(z)\mathbb{E}_{y|\mathbf{x}}\mathbb{1}_{\{z \geq y\}} + \mathbb{E}_{y|\mathbf{x}}\mathbb{1}_{\{z \geq y\}} dz  \tag{24}$$

$$= \mathbb{E}_{\mathbf{x}} \frac{1}{2} \int_{\mathbb{R}} F^2(z) - 2F(z)P_{y|\mathbf{x}}(z) + P_{y|\mathbf{x}}(z)dz.  \tag{25}$$

Here we used the law of total expectation and Fubini theorem to exchange the order of integration and then used the fact that $\mathbb{E}_{y|\mathbf{x}}\mathbb{1}_{\{z \geq y\}} = P_{y|\mathbf{x}}(z)$. Completing the square we further get:

$$\mathbb{E}\,\rho(y, F^{-1}(q)) = \mathbb{E}_{\mathbf{x}} \frac{1}{2} \int_{\mathbb{R}} F^2(z) - 2F(z)P_{y|\mathbf{x}}(z) + P_{y|\mathbf{x}}(z) + P_{y|\mathbf{x}}^2(z) - P_{y|\mathbf{x}}^2(z)dz  \tag{26}$$

$$= \mathbb{E}_{\mathbf{x}} \frac{1}{2} \int_{\mathbb{R}} (F(z) - P_{y|\mathbf{x}}(z))^2 + P_{y|\mathbf{x}}(z) - P_{y|\mathbf{x}}^2(z)dz  \tag{27}$$

$F = P_{y|\mathbf{x}}$ is clearly the unique minimizer of the last expression since $\int_{\mathbb{R}} (F(z) - P_{y|\mathbf{x}}(z))^2 dz > 0, \forall F \neq P_{y|\mathbf{x}}$. $\qquad \square$

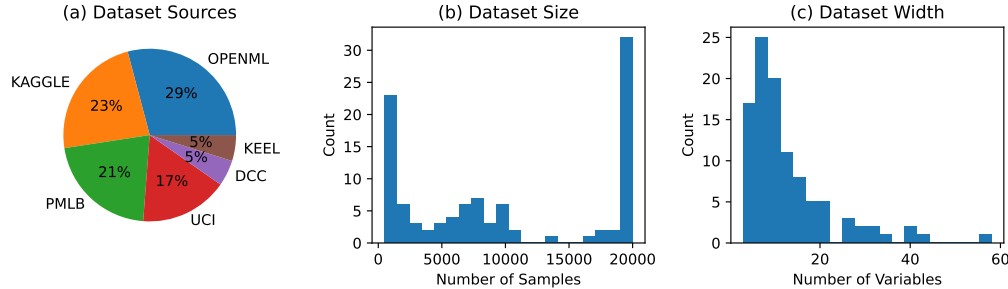

Figure 4: Summary statistics of the LPRM-101 benchmark. (a) The distribution by dataset sources, (b) the distribution of dataset sizes, (c) the distribution of variable count per dataset.

## B  LPRM-101 BENCHMARK DETAILS

LPRM-101 is the multi-dataset benchmark for large probabilistic regression models (hence, LPRM) consisting of 101 dataset (hence LPRM-101). The datasets, along with their sample count, number of variables and source information are listed in Table 3. To construct the benchmark, we first collect 101 dataset publicly available from the following primary repositories: UCI (Kelly et al., 2017), Kaggle (Kaggle, 2024), PMLB (Romano et al., 2021; Olson et al., 2017), OpenML (Vanschoren et al., 2013), KEEL (Alcalá-Fdez et al., 2011). We focus specifically on the regression task in which the dependent variable is continuous or, if it has limited number of levels, these are ordered such as student exam scores or wine quality. The target variable in each dataset is normalized to the [0, 10] range and the independent variables are used as is, raw. The target variable scaling is applied to equalize the contributions of the evaluation metrics from each dataset. Datasets have variable number of samples, the lowest being just below 1000. For very large datasets we limit the number of samples used in our benchmark to be 20,000 by subsampling uniformly at random. This allows us (i) to model imbalance, and at the same time (ii) avoid the situation in which a few large datasets could completely dominate the training and evaluation of the model. The distribution of datasets by source, number of samples and number of variables is shown in Figure 4.

For evaluating the prediction accuracy we use the following point prediction accuracy metrics: MAPE, SMAPE, AAD, RMSE, BIAS and distributional prediction accuracy metrics: CRPS and COVERAGE. We implement the 0.8/0.1/0.1 training/validation/test split sampled uniformly at random. Evaluation metrics are averaged over all samples in the test split containing samples from all datasets. The ground truth sample is denoted as $y_i$ and it's $q$-th quantile prediction as $\hat{y}_{i,q}$. Given the $N$-sample dataset, the point prediction accuracy metrics are defined as:

$$\text{SMAPE} = \frac{200}{N} \sum_{i=1}^{N} \frac{|y_i - \hat{y}_{i,0.5}|}{|y_i| + |\hat{y}_{i,0.5}|} \tag{28}$$

$$\text{MAPE} = \frac{100}{N} \sum_{i=1}^{N} \frac{|y_i - \hat{y}_{i,0.5}|}{|y_i|} \tag{29}$$

$$\text{AAD} = \frac{1}{N} \sum_{i=1}^{N} |y_i - \hat{y}_{i,0.5}| \tag{30}$$

$$\text{RMSE} = \sqrt{\frac{1}{N} \sum_{i=1}^{N} (y_i - \hat{y}_{i,0.5})^2} \tag{31}$$

$$\text{BIAS} = \frac{1}{N} \sum_{i=1}^{N} \hat{y}_{i,0.5} - y_i \tag{32}$$

The distributional accuracy metrics are defined over a random set of $Q = 200$ quantiles sampled uniformly at random and are formally defined as follows:

$$\text{CRPS} = \frac{1}{NQ} \sum_{i=1}^{N} \sum_{j=1}^{Q} \rho(y_i, \hat{y}_{i,q_j}), \tag{33}$$

$$\text{COVERAGE} @ q = \frac{100}{N} \sum_{i=1}^{N} \mathbb{1}[y_i > \hat{y}_{i,0.5-\alpha/200}] \mathbb{1}[y_i < \hat{y}_{i,0.5+\alpha/200}]. \tag{34}$$

Table 3: The list of dat

| | name | n_samples | n_vars | source | url |
|---|---|---|---|---|---|
| 0 | Abalone | 4177 | 7 | uci | `https://archive.ics.uci.` |
| 1 | Student_Performance | 649 | 29 | uci | `https://archive.ics.uci.` |
| 2 | Infrared_Thermography_Temperature | 1020 | 32 | uci | `https://archive.ics.uci.` |
| 3 | Parkinsons_Telemonitoring | 5875 | 18 | uci | `https://archive.ics.uci.` |
| 4 | Energy_Efficiency | 768 | 7 | uci | `https://archive.ics.uci.` |
| 5 | 1027_ESL | 488 | 3 | pmlb | `https://github.com/Epist` |
| 6 | 1028_SWD | 1000 | 9 | pmlb | `https://github.com/Epist` |
| 7 | 1029_LEV | 1000 | 3 | pmlb | `https://github.com/Epist` |
| 8 | 1030_ERA | 1000 | 3 | pmlb | `https://github.com/Epist` |
| 9 | 1199_BNG_echoMonths | 17496 | 8 | pmlb | `https://github.com/Epist` |
| 10 | 197_cpu_act | 8192 | 20 | pmlb | `https://github.com/Epist` |
| 11 | 225_puma8NH | 8192 | 7 | pmlb | `https://github.com/Epist` |
| 12 | 227_cpu_small | 8192 | 11 | pmlb | `https://github.com/Epist` |
| 13 | 294_satellite_image | 6435 | 35 | pmlb | `https://github.com/Epist` |
| 14 | 344_mv | 20000 | 9 | pmlb | `https://github.com/Epist` |
| 15 | 503_wind | 6574 | 13 | pmlb | `https://github.com/Epist` |
| 16 | 529_pollen | 3848 | 3 | pmlb | `https://github.com/Epist` |
| 17 | 537_houses | 20000 | 7 | pmlb | `https://github.com/Epist` |
| 18 | 547_no2 | 500 | 6 | pmlb | `https://github.com/Epist` |
| 19 | 564_fried | 20000 | 9 | pmlb | `https://github.com/Epist` |
| 20 | 595_fri_c0_1000_10 | 1000 | 9 | pmlb | `https://github.com/Epist` |
| 21 | 593_fri_c1_1000_10 | 1000 | 9 | pmlb | `https://github.com/Epist` |
| 22 | 1193_BNG_lowbwt | 20000 | 8 | pmlb | `https://github.com/Epist` |
| 23 | 1201_BNG_breastTumor | 20000 | 8 | pmlb | `https://github.com/Epist` |
| 24 | 1203_BNG_pwLinear | 20000 | 9 | pmlb | `https://github.com/Epist` |
| 25 | 215_2dplanes | 20000 | 9 | pmlb | `https://github.com/Epist` |
| 26 | 218_house_8L | 20000 | 7 | pmlb | `https://github.com/Epist` |
| 27 | QsarFishToxicity | 908 | 5 | uci | `https://archive.ics.uci.` |
| 28 | CONCRETE_COMPRESSIVE_STRENGTH | 1030 | 7 | uci | `https://archive.ics.uci.` |
| 29 | PRODUCTIVITY | 1197 | 12 | uci | `https://archive.ics.uci.` |
| 30 | CCPP | 9568 | 3 | uci | `https://archive.ics.uci.` |
| 31 | AIRFOIL | 1503 | 4 | uci | `https://archive.ics.uci.` |
| 32 | TETOUAN | 20000 | 6 | uci | `https://archive.ics.uci.` |
| 33 | BIAS_CORRECTION | 7725 | 22 | uci | `https://archive.ics.uci.` |
| 34 | APARTMENTS | 10000 | 10 | uci | `https://archive.ics.uci.` |
| 35 | MedicalCost | 1338 | 5 | kaggle | `kaggledatasetsdownload-c` |
| 36 | Vehicle | 2059 | 18 | kaggle | `kaggledatasetsdownload-c` |
| 37 | LifeExpectancy | 2928 | 18 | kaggle | `kaggledatasetsdownload-c` |
| 38 | CalHousing | 20000 | 7 | dcc | `https://www.dcc.fc.up.pt` |
| 39 | Ailerons | 7154 | 39 | dcc | `https://www.dcc.fc.up.pt` |
| 40 | DeltaElevators | 9517 | 5 | dcc | `https://www.dcc.fc.up.pt` |
| 41 | Pole | 10000 | 25 | dcc | `https://www.dcc.fc.up.pt` |
| 42 | Kinematics | 8192 | 7 | dcc | `https://www.dcc.fc.up.pt` |
| 43 | BigMartSales | 8523 | 10 | kaggle | `kaggledatasetsdownload-c` |
| 44 | VideoGameSales | 16598 | 3 | kaggle | `kaggledatasetsdownload-c` |
| 45 | NewsPopularity | 20000 | 58 | uci | `https://archive.ics.uci.` |
| 46 | Wizmir | 1461 | 8 | keel | `https://sci2s.ugr.es/kee` |

Table 3: The list of dat

| | name | n_samples | n_vars | source | url |
|---|---|---|---|---|---|
| 47 | Ele2 | 1056 | 3 | keel | https://sci2s.ugr.es/kee |
| 48 | Treasury | 1049 | 14 | keel | https://sci2s.ugr.es/kee |
| 49 | Mortgage | 1049 | 14 | keel | https://sci2s.ugr.es/kee |
| 50 | Laser | 993 | 3 | keel | https://sci2s.ugr.es/kee |
| 51 | SpaceGa | 3107 | 5 | openml | https://www.openml.org/c |
| 52 | VisualizingSoil | 8641 | 3 | openml | https://www.openml.org/c |
| 53 | Diamonds | 20000 | 8 | openml | https://www.openml.org/c |
| 54 | TitanicFare | 1307 | 6 | openml | https://www.openml.org/c |
| 55 | Sulfur | 10081 | 5 | openml | https://www.openml.org/c |
| 56 | Debutanizer | 2394 | 6 | openml | https://www.openml.org/c |
| 57 | Fardamento | 6277 | 5 | openml | https://www.openml.org/c |
| 58 | ProteinTertiary | 20000 | 8 | openml | https://api.openml.org/c |
| 59 | BrazilianHouses | 10692 | 7 | openml | https://api.openml.org/c |
| 60 | Cps88Wages | 20000 | 5 | openml | https://api.openml.org/c |
| 61 | CPMP-2015 | 2108 | 25 | openml | https://www.openml.org/c |
| 62 | NASA-PHM2008 | 20000 | 16 | openml | https://www.openml.org/c |
| 63 | Wind | 6574 | 12 | openml | https://www.openml.org/c |
| 64 | NewFuelCar | 20000 | 17 | openml | https://www.openml.org/c |
| 65 | MiamiHousing | 13932 | 14 | openml | https://www.openml.org/c |
| 66 | BlackFriday | 20000 | 8 | openml | https://www.openml.org/c |
| 67 | IEEE80211aaGATS | 5296 | 28 | openml | https://www.openml.org/c |
| 68 | Yprop41 | 8885 | 41 | openml | https://api.openml.org/c |
| 69 | Sarcos | 20000 | 20 | openml | https://api.openml.org/c |
| 70 | ZurichDelays | 20000 | 16 | openml | https://www.openml.org/c |
| 71 | 1000-Cameras | 1015 | 13 | openml | https://www.openml.org/c |
| 72 | GridStability | 10000 | 11 | openml | https://api.openml.org/c |
| 73 | PumaDyn32nh | 8192 | 31 | openml | https://api.openml.org/c |
| 74 | Fifa | 19178 | 27 | openml | https://api.openml.org/c |
| 75 | WhiteWine | 4898 | 10 | openml | https://api.openml.org/c |
| 76 | RedWine | 1599 | 10 | openml | https://api.openml.org/c |
| 77 | FpsBenchmark | 20000 | 42 | openml | https://api.openml.org/c |
| 78 | KingCountyHousing | 20000 | 20 | openml | https://api.openml.org/c |
| 79 | AvocadoPrices | 18249 | 12 | kaggle | kaggledatasetsdownload-c |
| 80 | Transcoding | 20000 | 18 | uci | https://archive.ics.uci. |
| 81 | house_16H | 20000 | 15 | openml | https://www.openml.org/c |
| 82 | Sales | 10738 | 13 | openml | https://www.openml.org/c |
| 83 | WalmartSales | 6435 | 8 | kaggle | kaggledatasetsdownload-c |
| 84 | UsedCar | 6019 | 11 | kaggle | kaggledatasetsdownload-c |
| 85 | HouseRent | 4746 | 11 | kaggle | kaggledatasetsdownload-c |
| 86 | LaptopPrice | 1273 | 15 | kaggle | kaggledatasetsdownload-c |
| 87 | UberFare | 20000 | 8 | kaggle | kaggledatasetsdownload-c |
| 88 | Co2Emission | 7385 | 10 | kaggle | kaggledatasetsdownload-c |
| 89 | SongPopularity | 18835 | 12 | kaggle | kaggledatasetsdownload-c |
| 90 | Cars | 20000 | 8 | kaggle | kaggledatasetsdownload-c |
| 91 | GemstonePrice | 20000 | 8 | kaggle | kaggledatasetsdownload-c |
| 92 | LoanAmount | 20000 | 20 | kaggle | kaggledatasetsdownload-c |
| 93 | SaudiArabiaCars | 5507 | 10 | kaggle | kaggledatasetsdownload-c |
| 94 | GpuKernelPerformance | 20000 | 13 | kaggle | kaggledatasetsdownload-c |
| 95 | AmericanHousePrices | 20000 | 10 | kaggle | kaggledatasetsdownload-c |
| 96 | KindleBooks | 20000 | 12 | kaggle | kaggledatasetsdownload-c |
| 97 | BookSales | 1070 | 8 | kaggle | kaggledatasetsdownload-c |
| 98 | CapitalGain | 20000 | 12 | kaggle | kaggledatasetsdownload-c |
| 99 | MarketingCampaign | 2976 | 14 | kaggle | kaggledatasetsdownload-c |

Table 3: The list of dat

| | name | n_samples | n_vars | source | url |
|---|---|---|---|---|---|
| 100 | CampaignUplift | 2000 | 9 | kaggle | kaggledatasetsdownload-c |

## C   Results with Confidence Intervals

To save space, we present benchmarking results with confidence intervals here. All confidence intervals are obtained by aggregating the evaluation results over 4 runs with different random seeds.

Table 4: Distributional accuracy of the proposed NIAQUE approach compared to the tree-based baselines and Transformer on LPRM-101 benchmark. Smaller values for CRPS are better. COVERAGE @ 95 values closer to 95 are better. The results with 95% confidence intervals derived from 4 random seed runs

|  | CRPS | COVERAGE @ 95 |
|---|---|---|
| XGBoost-global | $0.636 \pm 0.165$ | $94.6 \pm 0.3$ |
| XGBoost-local | $0.334 \pm 0.001$ | $90.8 \pm 0.2$ |
| LightGBM-global | $0.426 \pm 0.017$ | $94.8 \pm 0.1$ |
| LightGBM-local | $0.327 \pm 0.001$ | $91.5 \pm 0.2$ |
| CATBOOST-global | $0.443 \pm 0.004$ | $94.9 \pm 0.2$ |
| CATBOOST-local | $0.315 \pm 0.001$ | $92.7 \pm 0.1$ |
| Transformer-global | $0.272 \pm 0.005$ | $94.6 \pm 0.3$ |
| NIAQUE-local | $0.267 \pm 0.011$ | $94.9 \pm 0.4$ |
| NIAQUE-global | $0.261 \pm 0.002$ | $94.6 \pm 0.2$ |

Table 5: Point prediction accuracy of the proposed NIAQUE approach compared to the tree-based baselines and Transformer on LPRM-101 benchmark. Smaller values for sMAPE, AAD, RMSE are better. BIAS values closer to zero are better. The results with 95% confidence intervals derived from 4 random seed runs.

|  | sMAPE | AAD | BIAS | RMSE |
|---|---|---|---|---|
| XGBoost-global | $31.4 \pm 4.4$ | $0.574 \pm 0.100$ | $-0.15 \pm 0.05$ | $1.056 \pm 0.143$ |
| XGBoost-local | $25.6 \pm 0.1$ | $0.433 \pm 0.001$ | $-0.03 \pm 0.01$ | $0.883 \pm 0.004$ |
| LightGBM-global | $27.5 \pm 0.1$ | $0.475 \pm 0.001$ | $-0.06 \pm 0.01$ | $0.930 \pm 0.003$ |
| LightGBM-local | $25.7 \pm 0.1$ | $0.427 \pm 0.003$ | $-0.03 \pm 0.01$ | $0.865 \pm 0.012$ |
| CATBOOST-global | $31.3 \pm 0.2$ | $0.561 \pm 0.006$ | $-0.12 \pm 0.02$ | $1.030 \pm 0.009$ |
| CATBoost-local | $24.3 \pm 0.1$ | $0.408 \pm 0.001$ | $-0.03 \pm 0.01$ | $0.840 \pm 0.003$ |
| Transformer-global | $23.1 \pm 0.3$ | $0.383 \pm 0.008$ | $-0.01 \pm 0.01$ | $0.806 \pm 0.015$ |
| NIAQUE-local | $22.8 \pm 0.4$ | $0.377 \pm 0.012$ | $-0.03 \pm 0.01$ | $0.797 \pm 0.019$ |
| NIAQUE-global | $22.1 \pm 0.1$ | $0.367 \pm 0.002$ | $-0.02 \pm 0.01$ | $0.787 \pm 0.005$ |

# D  XGBOOST BASELINE

Table 6: Ablation study of the XGBoost model.

| type | max depth | learning rate | sMAPE | AAD | BIAS | RMSE | CRPS | COVERAGE @ 95 |
|------|-----------|---------------|-------|-----|------|------|------|----------------|
| global | 8  | 0.02 | 31.4 | 0.574 | -0.15 | 1.056 | 0.636 | 94.6 |
| global | 16 | 0.02 | 25.7 | 0.441 | -0.07 | 0.864 | 0.484 | 91.5 |
| global | 32 | 0.02 | 24.1 | 0.402 | -0.05 | 0.800 | 0.353 | 80.0 |
| global | 40 | 0.02 | 24.6 | 0.414 | -0.05 | 0.815 | 0.378 | 78.2 |
| global | 48 | 0.02 | 24.1 | 0.397 | -0.04 | 0.785 | 0.362 | 74.8 |
| global | 96 | 0.02 | 23.8 | 0.384 | -0.03 | 0.769 | 0.346 | 64.9 |
| local | 16 | 0.02 | 23.0 | 0.367 | -0.00 | 0.753 | 0.317 | 52.0 |
| local | 12 | 0.02 | 22.7 | 0.369 | -0.01 | 0.756 | 0.304 | 66.0 |
| local | 8  | 0.02 | 22.4 | 0.372 | -0.02 | 0.773 | 0.294 | 82.3 |
| local | 8  | 0.05 | 22.5 | 0.373 | -0.02 | 0.773 | 0.291 | 82.4 |
| local | 6  | 0.02 | 22.7 | 0.382 | -0.02 | 0.795 | 0.298 | 87.3 |
| local | 4  | 0.02 | 24.1 | 0.412 | -0.03 | 0.847 | 0.318 | 90.2 |
| local | 3  | 0.02 | 25.6 | 0.433 | -0.03 | 0.883 | 0.334 | 90.8 |

# E CATBOOST BASELINE

The CATBoost is trained using the standard package via `pip install catboost` using `grow_policy = Depthwise`. The explored hyper-parqameter grid appears in Table 7.

Table 8 shows CATBoost accuracy as a function of the number of quantiles. Quantiles are generated using linspace grid `np.linspace(0.01, 0.99, num_quantiles)`. We recover the best overall result for the case of 3 quantiles, and increasing the number of quantiles leads to quickly deteriorating metrics. It appears that CATBoost is unfit to solve complex multi-quantile problems.

Table 7: Ablation study of the CATBoost model.

| type | depth | min data in leaf | SMAPE | AAD | BIAS | RMSE | CRPS | COVERAGE @ 95 |
|------|-------|------------------|-------|-----|------|------|------|----------------|
| global | 16 | 50 | 31.4 | 0.565 | -0.12 | 1.036 | 0.442 | 94.2 |
| global | 16 | 100 | 31.3 | 0.561 | -0.12 | 1.030 | 0.443 | 94.9 |
| global | 16 | 200 | 31.6 | 0.569 | -0.13 | 1.041 | 0.445 | 94.2 |
| global | 8 | 100 | 41.1 | 0.785 | -0.26 | 1.324 | 0.602 | 94.3 |
| local | 3 | 50 | 24.3 | 0.409 | -0.03 | 0.841 | 0.316 | 92.7 |
| local | 3 | 100 | 24.3 | 0.407 | -0.03 | 0.843 | 0.317 | 92.7 |
| local | 3 | 200 | 24.3 | 0.408 | -0.03 | 0.840 | 0.315 | 92.7 |
| local | 5 | 50 | 22.2 | 0.373 | -0.02 | 0.785 | 0.285 | 90.7 |
| local | 5 | 100 | 22.3 | 0.374 | -0.02 | 0.786 | 0.285 | 91.3 |
| local | 5 | 200 | 22.4 | 0.378 | -0.02 | 0.791 | 0.288 | 91.6 |
| local | 7 | 50 | 21.5 | 0.359 | -0.02 | 0.761 | 0.272 | 87.2 |
| local | 7 | 100 | 21.6 | 0.362 | -0.02 | 0.765 | 0.273 | 88.6 |
| local | 7 | 200 | 21.8 | 0.366 | -0.02 | 0.772 | 0.277 | 89.9 |

Table 8: CATBoost accuracy as a function of the number of quantiles.

| type | depth | min data in leaf | num quantiles | SMAPE | AAD | BIAS | RMSE | CRPS | COVERAGE @ 95 |
|------|-------|------------------|---------------|-------|-----|------|------|------|----------------|
| global | 16 | 100 | 3 | 31.3 | 0.561 | -0.12 | 1.030 | 0.443 | 94.9 |
| global | 16 | 100 | 5 | 35.0 | 0.665 | -0.13 | 1.183 | 0.482 | 96.2 |
| global | 16 | 100 | 7 | 38.5 | 0.746 | -0.18 | 1.265 | 0.533 | 96.2 |
| global | 16 | 100 | 9 | 43.7 | 0.879 | -0.25 | 1.437 | 0.622 | 96.2 |
| global | 16 | 100 | 51 | 68.9 | 1.538 | -0.53 | 2.132 | 1.036 | 95.5 |
| local | 7 | 100 | 3 | 21.5 | 0.359 | -0.02 | 0.761 | 0.272 | 87.2 |
| local | 7 | 100 | 9 | 23.9 | 0.399 | -0.03 | 0.823 | 0.284 | 92.4 |
| local | 7 | 100 | 51 | 30.3 | 0.525 | -0.09 | 1.079 | 0.369 | 92.1 |
| local | 16 | 100 | 51 | 30.2 | 0.514 | -0.09 | 1.055 | 0.362 | 92.4 |

# F LightGBM Baseline

Table 9: Ablation study of the LightGBM model.

| type | max_depth | num leaves | learning rate | SMAPE | AAD | BIAS | RMSE | CRPS | COVERAGE @ 95 |
|---|---|---|---|---|---|---|---|---|---|
| global | -1 | 10 | 0.05 | 35.6 | 0.661 | -0.17 | 1.199 | 0.804 | 95.2 |
| global | -1 | 20 | 0.05 | 30.9 | 0.554 | -0.11 | 1.034 | 0.566 | 95.4 |
| global | -1 | 40 | 0.05 | 27.5 | 0.475 | -0.06 | 0.930 | 0.426 | 94.8 |
| global | -1 | 100 | 0.05 | 24.6 | 0.417 | -0.03 | 0.852 | 0.342 | 93.3 |
| global | -1 | 200 | 0.05 | 23.4 | 0.393 | -0.02 | 0.813 | 0.32 | 92.3 |
| global | -1 | 400 | 0.05 | 23.6 | 0.379 | -0.02 | 0.786 | 0.305 | 90.9 |
| global | 3 | 10 | 0.05 | 50.7 | 1.084 | -0.49 | 1.763 | 1.013 | 94.1 |
| global | 3 | 20 | 0.05 | 50.7 | 1.084 | -0.49 | 1.763 | 1.013 | 94.1 |
| global | 3 | 40 | 0.05 | 50.7 | 1.084 | -0.49 | 1.763 | 1.013 | 94.1 |
| global | 3 | 100 | 0.05 | 50.7 | 1.084 | -0.49 | 1.763 | 1.013 | 94.1 |
| global | 3 | 200 | 0.05 | 50.7 | 1.084 | -0.49 | 1.763 | 1.013 | 94.1 |
| global | 3 | 400 | 0.05 | 50.7 | 1.084 | -0.49 | 1.763 | 1.013 | 94.1 |
| global | 5 | 10 | 0.05 | 39.1 | 0.768 | -0.25 | 1.341 | 0.856 | 94.8 |
| global | 5 | 20 | 0.05 | 39.0 | 0.76 | -0.26 | 1.327 | 0.863 | 94.8 |
| global | 5 | 40 | 0.05 | 39.0 | 0.759 | -0.26 | 1.328 | 0.864 | 94.8 |
| global | 5 | 100 | 0.05 | 39.0 | 0.759 | -0.26 | 1.328 | 0.864 | 94.8 |
| global | 5 | 200 | 0.05 | 39.0 | 0.759 | -0.26 | 1.328 | 0.864 | 94.8 |
| global | 5 | 400 | 0.05 | 39.0 | 0.759 | -0.26 | 1.328 | 0.864 | 94.8 |
| global | 10 | 10 | 0.05 | 35.6 | 0.661 | -0.17 | 1.199 | 0.804 | 95.2 |
| global | 10 | 20 | 0.05 | 31.5 | 0.572 | -0.14 | 1.054 | 0.59 | 95.4 |
| global | 10 | 40 | 0.05 | 29.8 | 0.537 | -0.13 | 1.001 | 0.575 | 95.2 |
| global | 10 | 100 | 0.05 | 29.5 | 0.528 | -0.12 | 0.991 | 0.577 | 95.2 |
| global | 10 | 200 | 0.05 | 29.2 | 0.522 | -0.12 | 0.981 | 0.576 | 95.0 |
| global | 10 | 400 | 0.05 | 29.1 | 0.52 | -0.12 | 0.975 | 0.582 | 95.1 |
| global | 20 | 10 | 0.05 | 35.6 | 0.661 | -0.17 | 1.199 | 0.804 | 95.2 |
| global | 20 | 20 | 0.05 | 30.9 | 0.554 | -0.11 | 1.034 | 0.566 | 95.4 |
| global | 20 | 40 | 0.05 | 27.1 | 0.468 | -0.07 | 0.913 | 0.512 | 95.2 |
| global | 20 | 100 | 0.05 | 25.5 | 0.435 | -0.06 | 0.864 | 0.496 | 94.9 |
| global | 20 | 200 | 0.05 | 25.0 | 0.424 | -0.06 | 0.846 | 0.488 | 94.3 |
| global | 20 | 400 | 0.05 | 24.3 | 0.41 | -0.05 | 0.823 | 0.482 | 93.6 |
| global | 40 | 10 | 0.05 | 35.6 | 0.661 | -0.17 | 1.199 | 0.804 | 95.2 |
| global | 40 | 20 | 0.05 | 30.9 | 0.554 | -0.11 | 1.034 | 0.566 | 95.4 |
| global | 40 | 40 | 0.05 | 27.8 | 0.481 | -0.05 | 0.913 | 0.431 | 94.7 |
| global | 40 | 100 | 0.05 | 24.7 | 0.419 | -0.04 | 0.848 | 0.348 | 93.5 |
| global | 40 | 200 | 0.05 | 23.5 | 0.395 | -0.03 | 0.811 | 0.332 | 92.7 |
| global | 40 | 400 | 0.05 | 23.2 | 0.383 | -0.03 | 0.791 | 0.322 | 92.0 |

Table 10: Ablation study of the LightGBM model.

| type | max_depth | num leaves | learning rate | sMAPE | AAD | BIAS | RMSE | CRPS | COVERAGE @ 95 |
|------|-----------|-----------|---------------|-------|-----|------|------|------|---------------|
| local | -1 | 5 | 0.05 | 23.8 | 0.399 | -0.03 | 0.823 | 0.319 | 90.6 |
| local | -1 | 10 | 0.05 | 22.5 | 0.376 | -0.02 | 0.786 | 0.301 | 88.9 |
| local | -1 | 20 | 0.05 | 21.9 | 0.364 | -0.02 | 0.766 | 0.289 | 86.5 |
| local | -1 | 50 | 0.05 | 21.6 | 0.355 | -0.01 | 0.752 | 0.278 | 82.6 |
| local | 2 | 5 | 0.05 | 25.7 | 0.427 | -0.03 | 0.865 | 0.327 | 91.5 |
| local | 2 | 10 | 0.05 | 25.7 | 0.427 | -0.03 | 0.865 | 0.327 | 91.5 |
| local | 2 | 20 | 0.05 | 25.7 | 0.427 | -0.03 | 0.865 | 0.327 | 91.5 |
| local | 2 | 50 | 0.05 | 25.7 | 0.427 | -0.03 | 0.865 | 0.327 | 91.5 |
| local | 3 | 5 | 0.05 | 24.3 | 0.404 | -0.03 | 0.83 | 0.318 | 90.7 |
| local | 3 | 10 | 0.05 | 23.9 | 0.396 | -0.03 | 0.818 | 0.304 | 90.4 |
| local | 3 | 20 | 0.05 | 23.9 | 0.396 | -0.03 | 0.818 | 0.304 | 90.4 |
| local | 3 | 50 | 0.05 | 23.9 | 0.396 | -0.03 | 0.818 | 0.304 | 90.4 |
| local | 5 | 5 | 0.05 | 23.8 | 0.399 | -0.03 | 0.823 | 0.319 | 90.6 |
| local | 5 | 10 | 0.05 | 22.7 | 0.379 | -0.02 | 0.79 | 0.3 | 89.1 |
| local | 5 | 20 | 0.05 | 22.3 | 0.37 | -0.02 | 0.776 | 0.287 | 87.6 |
| local | 5 | 50 | 0.05 | 22.2 | 0.368 | -0.02 | 0.773 | 0.285 | 87.4 |

# G  TRANSFORMER BASELINE

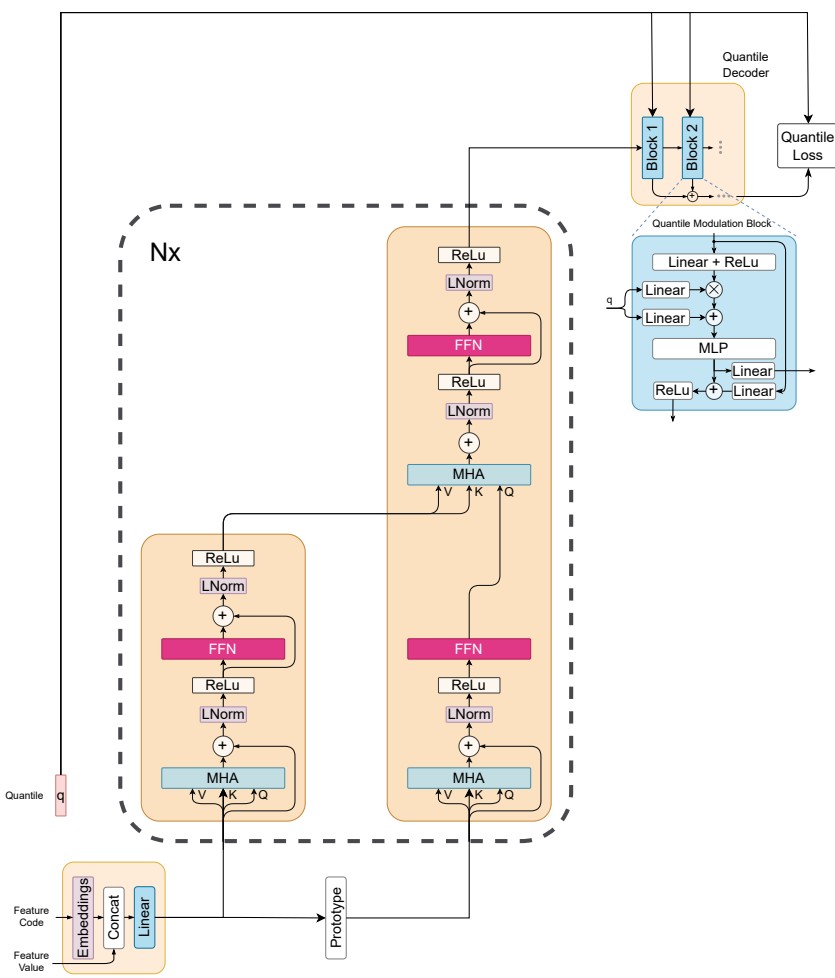

Figure 5: Transformer baseline used in our experiments. The feature encoding module is replaced with transformer block. Feature encoding is implemented via self-attention. The extraction of feature encoding is done by applying cross-attention between the prototype of input features and the output of self-attention. This operation is repeated several times corresponding to the number of blocks in transformer encoder.

The ablation study of the transformer architecture is presented in Table 11. It shows that in general, increasing the number of transformer blocks improves accuracy, however, at 8-10 blocks we clearly see diminishing returns. Dropout helps to gain better empirical coverage of the 95% confidence interval, but this happens at the expense of point prediction accuracy. Finally, the decoder query that is used to produce the feature embedding that is fed to the quantile decoder can be implemented in two principled ways. First, the scheme depicted in Figure 5, uses the prototype of features supplied to the encoder. We call it the prototype scheme. Second, the prototype can be replaced by a learnable embedding. Comparing the last and third rows in Table 11, we conclude that the prototype scheme is a clear winner.

Table 11: Ablation study of the Transformer architecture.

| query | d_model | width | blocks | dp | SMAPE | AAD | BIAS | RMSE | CRPS | COVERAGE @ 95 |
|-------|---------|-------|--------|-----|-------|-------|-------|-------|-------|----------------|
| proto | 256 | 256 | 4 | 0.1 | 25.6 | 0.462 | -0.01 | 0.918 | 0.313 | 95.2 |
| proto | 256 | 1024 | 4 | 0.1 | 24.5 | 0.414 | -0.02 | 0.845 | 0.292 | 95.1 |
| proto | 256 | 256 | 6 | 0.1 | 23.7 | 0.397 | -0.01 | 0.824 | 0.281 | 94.9 |
| proto | 256 | 512 | 6 | 0.2 | | | | | | |
| proto | 256 | 1024 | 6 | 0.1 | 24.3 | 0.407 | -0.01 | 0.840 | 0.287 | 94.9 |
| proto | 256 | 1024 | 6 | 0.0 | 26.5 | 0.477 | -0.04 | 0.980 | 0.334 | 93.0 |
| proto | 256 | 512 | 8 | 0.0 | 23.3 | 0.388 | -0.03 | 0.814 | 0.276 | 94.3 |
| proto | 256 | 1024 | 8 | 0.0 | 23.1 | 0.383 | -0.02 | 0.806 | 0.272 | 94.6 |
| proto | 256 | 1024 | 8 | 0.1 | 23.1 | 0.384 | -0.01 | 0.809 | 0.272 | 94.6 |
| proto | 256 | 512 | 10 | 0.0 | 23.0 | 0.384 | -0.03 | 0.814 | 0.273 | 94.2 |
| proto | 256 | 1024 | 10 | 0.1 | 24.3 | 0.407 | -0.01 | 0.840 | 0.287 | 94.9 |
| proto | 512 | 1024 | 6 | 0.1 | | | | | | |
| learn | 256 | 256 | 6 | 0.2 | 35.0 | 0.722 | -0.16 | 1.406 | 0.489 | 93.9 |

# H   NIAQUE-LOCAL BASELINE

NIAQUE-local baseline is trained on each dataset individually using the same overall training framework as discussed in the main manuscript for the NIAQUE-global, with the following exceptions. The number of training epochs for each dataset is fixed at 1200, the batch size is set to 256, feature dropout is disabled. Finally, for each dataset we select the best model to be evaluated by monitoring the loss on validation set every epoch.

Table 12: Ablation study of NIAQUE-local model.

| blocks | width | dp | layers | SMAPE | AAD | BIAS | RMSE | CRPS | COVERAGE @ 95 |
|--------|-------|-----|--------|-------|-------|-------|-------|-------|---------------|
| 2 | 64 | 0.0 | 3 | 24.2 | 0.414 | -0.03 | 0.848 | 0.292 | 95.1 |
| 2 | 128 | 0.0 | 3 | 22.8 | 0.381 | -0.02 | 0.804 | 0.270 | 94.5 |
| 2 | 256 | 0.0 | 3 | 22.1 | 0.365 | -0.02 | 0.786 | 0.260 | 94.0 |
| 2 | 512 | 0.0 | 3 | 21.9 | 0.360 | -0.02 | 0.781 | 0.257 | 92.7 |
| 2 | 64 | 0.1 | 3 | 24.7 | 0.431 | -0.07 | 0.855 | 0.305 | 93.3 |
| 2 | 128 | 0.1 | 3 | 23.1 | 0.389 | -0.04 | 0.81 | 0.276 | 94.0 |
| 2 | 256 | 0.1 | 3 | 22.2 | 0.369 | -0.02 | 0.79 | 0.263 | 94.0 |
| 2 | 512 | 0.1 | 3 | 22.0 | 0.361 | -0.02 | 0.779 | 0.257 | 93.5 |
| 2 | 64 | 0.0 | 2 | 24.5 | 0.419 | -0.03 | 0.852 | 0.296 | 95.0 |
| 2 | 128 | 0.0 | 2 | 23.4 | 0.391 | -0.02 | 0.815 | 0.276 | 94.7 |
| 2 | 256 | 0.0 | 2 | 22.3 | 0.368 | -0.02 | 0.783 | 0.262 | 94.1 |
| 2 | 512 | 0.0 | 2 | 22.1 | 0.363 | -0.03 | 0.780 | 0.259 | 92.9 |
| 4 | 64 | 0.0 | 2 | 23.8 | 0.399 | -0.02 | 0.828 | 0.282 | 95.1 |
| 4 | 128 | 0.0 | 2 | 22.8 | 0.377 | -0.03 | 0.797 | 0.267 | 94.9 |
| 4 | 256 | 0.0 | 2 | 22.0 | 0.363 | -0.02 | 0.788 | 0.259 | 93.5 |
| 4 | 512 | 0.0 | 2 | 22.0 | 0.359 | -0.02 | 0.785 | 0.257 | 92.0 |
| 4 | 64 | 0.1 | 2 | 23.8 | 0.401 | -0.03 | 0.829 | 0.284 | 94.3 |
| 4 | 128 | 0.1 | 2 | 22.9 | 0.379 | -0.03 | 0.801 | 0.267 | 94.6 |
| 4 | 256 | 0.1 | 2 | 22.1 | 0.363 | -0.03 | 0.786 | 0.259 | 93.5 |
| 4 | 512 | 0.1 | 2 | 22.0 | 0.360 | -0.03 | 0.781 | 0.257 | 92.4 |
| 8 | 128 | 0.0 | 2 | 23.0 | 0.381 | -0.02 | 0.798 | 0.27 | 95.7 |

Table 13: Ablation study of NIAQUE model.

| blocks | width | dp | layers | singles | log input | SMAPE | AAD | BIAS | RMSE | CRPS | COVERAGE @ 95 |
|--------|-------|-----|--------|---------|-----------|-------|-------|-------|-------|-------|---------------|
| 1 | 1024 | 0.2 | 2 | 5% | yes | 25.6 | 0.433 | -0.04 | 0.864 | 0.306 | 96.5 |
| 2 | 1024 | 0.2 | 2 | 5% | yes | 23.1 | 0.384 | -0.02 | 0.802 | 0.272 | 95.7 |
| 2 | 1024 | 0.2 | 3 | 5% | yes | 22.7 | 0.377 | -0.03 | 0.796 | 0.267 | 95.6 |
| 4 | 1024 | 0.2 | 2 | 5% | yes | 22.1 | 0.367 | -0.02 | 0.787 | 0.261 | 94.6 |
| 4 | 1024 | 0.2 | 3 | 5% | yes | 22.1 | 0.367 | -0.02 | 0.792 | 0.262 | 94.6 |
| 8 | 1024 | 0.2 | 2 | 5% | yes | 22.0 | 0.366 | -0.02 | 0.798 | 0.264 | 92.7 |
| 4 | 512 | 0.2 | 2 | 0% | yes | 22.5 | 0.372 | -0.02 | 0.791 | 0.264 | 95.4 |
| 4 | 1024 | 0.2 | 2 | 0% | yes | 22.1 | 0.366 | -0.02 | 0.791 | 0.261 | 94.2 |
| 4 | 1024 | 0.3 | 2 | 0% | yes | 22.1 | 0.367 | -0.02 | 0.787 | 0.260 | 94.7 |
| 4 | 1024 | 0.4 | 2 | 0% | yes | 22.2 | 0.370 | -0.02 | 0.791 | 0.263 | 95.1 |
| 4 | 2048 | 0.3 | 2 | 0% | yes | 22.1 | 0.366 | -0.02 | 0.795 | 0.263 | 93.4 |
| 4 | 1024 | 0.2 | 2 | 5% | no | 31.4 | 0.530 | -0.066 | 1.017 | 0.371 | 95.6 |

# I  NIAQUE TRAINING DETAILS AND ABLATION STUDIES

To train both NIAQUE and Transformer models we use feature dropout defined as follows. Given dropout probability dp, we toss a coin with probability $\sqrt{dp}$ to determine if the dropout event is going to happen at all for a given batch. If this happens, we remove each feature from the batch, again with probability $\sqrt{dp}$. This way each feature has probability dp of being removed from a given batch and there is a probability $\sqrt{dp}$ that the model will see all features intact in a given batch. The intuition behind this design is that we want to expose the model to all features most of the time, but we also want to create many situations with some feature combinations missing.

Architecture and training ablations are reported in Table 13 shown that increasing the number of blocks and width improves accuracy until saturation happens at 4 blocks and width 1024.

**Input log transformation** defined in eq. (8) is important to ensure the success of the training, as follows both from Table 13 and Figure 6. The introduction of log-transform makes learning curves well-behaved and smooth and translates into much better accuracy.

**Adding samples containing only one of the features** as input does not significantly affect accuracy. At the same time, the addition of single-feature training rows has very strong effect on the effectiveness of NIAQUE's interpretability mechanism. When rows with single feature input are added (Figures 7a and 7b), NIAQUE demonstrates very clear accuracy degradation when top features are removed and insignificant degradation when bottom features are removed. When rows with single feature input are *not* added (Figure 7c), the discrimination between strong and weak features is poor, with removal of top and bottom features having approximately the same effect across datasets.

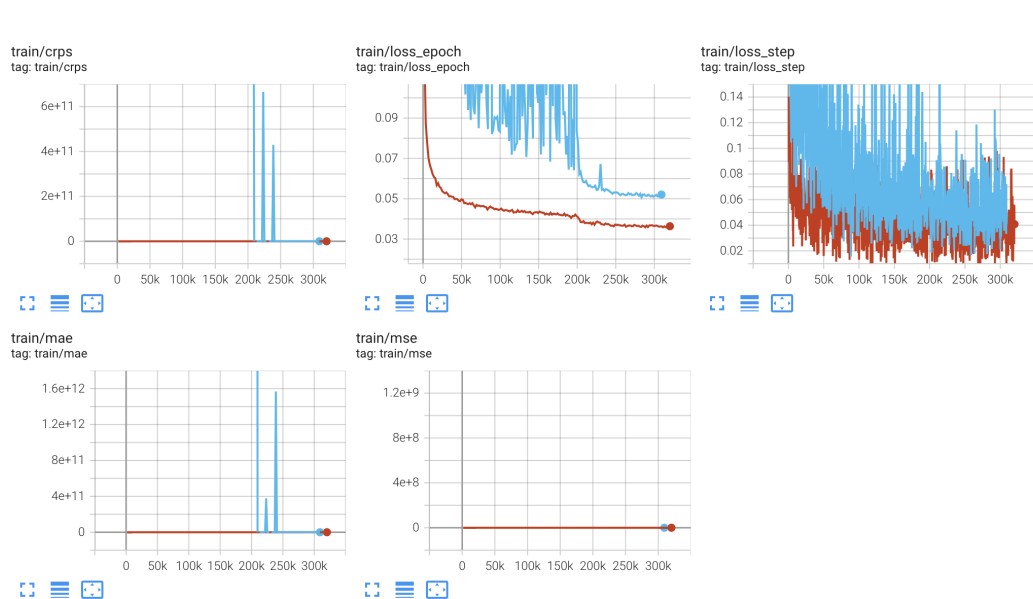

Figure 6: Training losses with (dark red) and without (blue) input value log-transform eq. (8). The introduction of log-transform makes learning curves well-behaved and smooth.

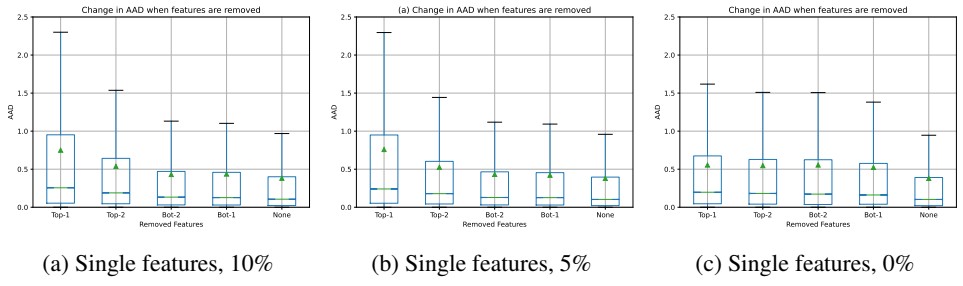

(a) Single features, 10%        (b) Single features, 5%        (c) Single features, 0%

Figure 7: The effect of adding training rows containing only one of the input features as NIAQUE input. When rows with single feature input are added (Figures 7a and 7b), NIAQUE demonstrates very clear accuracy degradation when top features are removed and insignificant degradation when bottom features are removed. When rows with single feature input are *not* added (Figure 7c), the discrimination between strong and weak features is poor, with removal of top and bottom features having approximately the same effect across datasets.

