# OpenReview forum: "NIAQUE: Neural Interpretable Any-Quantile Estimation - Towards Large Probabilistic Regression Models"
_ICLR.cc/2025/Conference — Submitted to ICLR 2025_

### Official Review · Reviewer_JSss · 2024-11-02

**Soundness:** 2
**Presentation:** 3
**Contribution:** 2
**Rating:** 5
**Confidence:** 3

**Summary:**

This paper aims to narrow the gap between deep probabilistic regression models with current powerful deep neural networks. It proposes a large unified model to handle regression tasks across different datasets. Correspondingly, it introduces a framework called NIAQUE, to address probabilistic regression problem. Compared with several baseline methods, the proposed model achieves better empirical results.

**Strengths:**

1. This paper aims to narrow the gap between current neural networks and probability regression models, which is still under explored.
2. It proposes a complete working pipeline from building benchmark, proposing new model with theoritical analysis, and empirical analysis.
3. Overall, the draft is easy to read and follow.

**Weaknesses:**

1. The problem statement is still not clear, why use transformer model to handle such tasks, what kind of real-world applications to illustrate the model practices? more discussions and intuitions are needed to support the motivation of this work.
2. Some table and figure formats are not well prepared such as figure 3. Necessary descriptions are needed.
3. Based on the table 1, there is not clear difference of final results compared with previous works, what is the benefit of proposing such a learning pipeline? how to illustrate the proposed model superiority?

**Questions:**

Please refer to the weaknesses section above.

---

> ### Author Response · Authors · 2024-11-25
> **Response**
>
> Dear Reviewer JSss, we would like to thank you for providing insightful feedback. Our detailed response appears below.
>
> 1. The Transformer model is used as a baseline. The basic principle here is that in order to operate in the multi-task setting, where each dataset represents a different task and has a different number of available features, the model needs to be able to process variable sized input. Both the proposed NIAQUE model and the Transformer model are capable of this mode operation, with NIAQUE demonstrating improved accuracy. Additionally, some of the existing distributional models such as neural process or conditional neural process pointed out by Reviewer p8Dt simply cannot do it. The multi-task operation creates the opportunity to co-train on multiple datasets and hence have one model for multiple problems. Finally, in our updated experiments, we show the feasibility of pretraining a large probabilistic regression model that can be fine-tuned on unseen target datasets, providing clear performance lift. We have revised the manuscript and included additional experiments and motivation for our work accordingly.
>
> 2. We have provided additional clarifications in the caption of Figure 3. We are happy to provide additional clarifications if Reviewer provides additional guidance.
>
> 3. We have provided additional transfer learning results in Table 2 and paragraph Transfer Learning Experiment. The learning pipeline we proposed provides the ability to train foundational probabilistic regression models that can be fine-tuned on target datasets providing a clear accuracy lift compared to the model trained on the same data from scratch. This is akin to the case of training foundational vision or language models that can be used in various downstream tasks through transfer learning. The feasibility of such transfer learning has not been demonstrated before in the context of probabilistic regression models and we are closing this gap in this paper.

---

### Official Review · Reviewer_NJbn · 2024-11-02

**Soundness:** 3
**Presentation:** 2
**Contribution:** 3
**Rating:** 6
**Confidence:** 4

**Summary:**

The current manuscript introduces NIAQUE, a neural network model for probabilistic regression that performs any-quantile estimation across multiple datasets.  NIAQUE takes a training batch of the variable number of observations and generates a quantile of uniformly at random in the [0,1] range for each training sample. The quantile is used as the input of the quantile decoder to modulate
the observation representation and as the supervision signal in the quantile loss.

Besides this, the paper also introduces the LPRM-101 Benchmark which is a new benchmark with 101 diverse regression datasets from different sources like UCI, PMLB, OpenML, and Kaggle, aimed at assessing the performance of models on probabilistic regression tasks. The proposed method enables to handle a range of quantiles in a single framework. Its encoder-decoder design effectively manages variable data formats across datasets, making it flexible for multi-dataset.

**Strengths:**

* The authors provided the LPRM-101 benchmark which allows for rigorous evaluation across diverse regression tasks and data sources.

* The proposed NIAQUE seems a new probabilistic regression that learns to approximate the inverse of the posterior distribution during training.

**Weaknesses:**

* W1: co-training across datasets could introduce biases that are absent in single-dataset models. This is especially relevant if certain datasets dominate the training process, potentially leading to skewed representations and reduced performance on minority datasets. It would be interesting to see the performance of your model in individual datasets when trained jointly vs. separately.

* W2: While the any-quantile framework is flexible, the performance of the model may vary depending on the choice of quantiles.The lack of specific guidance on the choice of quantiles may prevent users from achieving optimal results in different use cases.

* W3:NIAQUE relies on specific probabilistic assumptions for feature importance and distributional modeling, which may not be suitable for all regression tasks, particularly those with non-standard distributional characteristics. This could limit the model’s applicability to a narrower range of tasks than intended.

**Questions:**

* Q1) I am wondering how NIAQUE's performance varies with different quantile choices.

---

> ### Author Response · Authors · 2024-11-25
> **Response**
>
> Dear Reviewer NJbn, we would like to thank you for providing insightful feedback. Our detailed response appears below.
>
> - W1: Indeed we have outlined this as a potential risk in the discussion section of our paper. We would like to note that this is not exclusive to our model, but rather is a common risk relevant for the class of multi-task models as a whole. We would also like to emphasize that most of the prominent present day models are multi-task models. We believe that an effective strategy to combat this risk factor is fine-tuning on the specific target dataset. In the light of our most recent transfer learning results summarized in Table 2 of the revised paper, we conclude that fine-tuning of a pretrained NIAQUE model is viable and produces tangible overall accuracy lift compared to the control model trained from scratch, even when we vary the size of the training set on the target unseen dataset, as shown in Table 2. This provides evidence that the model is capable of producing improved results for target datasets of varying sizes, addressing concerns raised in this comment. Additionally, based on our preliminary analysis, it appears that the lift is consistent across datasets. The metric comparisons on individual datasets for pretrained/global models and the local model will be presented in an appendix in the camera ready version of the paper.
>
> - W2: NIAQUE model provides the capability to learn arbitrary quantiles of the target distribution. As such, all metrics of the model remain the same irrespective of the quantiles selected to compute COVERAGE @ $\alpha$ metric. The only metric that changes is COVERAGE @ $\alpha$. This is because (i) the point estimation metrics are computed based on the median and (ii) the CRPS metric is computed based on 200 quantiles sampled uniformly at random in (0, 1). In this light, the CRPS metric already does provide a measure of model success when operating on arbitrary quantiles, not just ones used for COVERAGE @ $\alpha$ computation. Additionally, for the NIAQUE model reported in Table 1, COVERAGE @ 50 is 52.2, COVERAGE @ 90 is 89.4 and COVERAGE @ 99 is 98.1. Note that this computation is done on the same model, without retraining it in any way for the specific target quantiles, but rather querying it with the user requested quantile value of interest. Therefore, the guideline for the user is as simple as: train the model in the way described in the paper and at inference time, query the model with the quantile of your choice, be it p50 (median), p5, p90 or any other. We have included the following clarification at the beginning of Section 2.1: "Therefore, at inference time the user of the model has the choice of querying the model with any combination of target quantiles that best suit the user's downstream application."
>
> - W3: Our model and training method do not make any parametric assumptions about the nature of the target distribution. Hence, it is able to handle distributions of different properties, naturally. The LPRM-101 benchmark is expansive and it contains datasets of very different nature. Figure 3b demonstrates what happens when features are ranked by the importance weight, calculated per dataset, and then removed from model input based on their importance rank. The low-importance feature removal degrades accuracy visibly less than the removal of high importance features, across many datasets. This clearly demonstrates that the proposed importance works on a wide variety of data. We believe that this addresses the concern raised by the reviewer.

---

> > ### Comment · Reviewer_NJbn · 2024-11-28
> > **response to authors rebuttal**
> >
> > I have carefully considered the feedback from other reviewers, as well as the author's response to neural process questions, and their responses to other reviewers. While I appreciate the authors' thorough and thoughtful responses, I am convinced by the explanations provided, and will slightly change my score to marginal acceptance! good luck!
> >
> > Moreover, I would encourage the authors to share the code [and datasets] while submitting a manuscript.

---

> > > ### Author Response · Authors · 2024-11-28
> > >
> > > Dear Reviewer NJbn, thank you very much!

---

### Official Review · Reviewer_p8Dt · 2024-11-04

**Soundness:** 3
**Presentation:** 3
**Contribution:** 2
**Rating:** 5
**Confidence:** 4

**Summary:**

The authors focus on the problem of multi-task probabilistic regression over tabular datasets. They contribute a benchmark of 101 regression datasets drawn from public sources, and develop a neural network architecture trained with quantile regression for this setting. They demonstrate that their architecture, NIAQUE, outperforms tree-based and Transformer baselines in both point and distributional metrics. The authors also provide a theoretical result demonstrating the consistency of their any-quantile training algorithm.

**Strengths:**

* Good motivation: The problem setting of multi-task learning across tabular regression datasets is a promising and important one, so the paper is well-motivated.
* NIAQUE compares reasonably well to strongest baseline: The NIAQUE architecture appears to have decent empirical results, performing slightly better than the Transformer with ~2x reduction in wall-clock training time (L254).
* Strong benchmark and set of evaluation datasets: The contributed LPRM-101 benchmark is expansive.
* The method (the neural network architecture and quantile regression objective) seem sensible and novel.
* The consistency result for NIAQUE appears to be correct.

**Weaknesses:**

* Minimal gains from multi-task transfer: the core pitch of the paper is to contribute an architecture that works well for multi-task learning, but the gains from learning across datasets appear to be pretty marginal (comparing NIAQUE-local and NIAQUE-global in Table 1).
* Unclear presentation: Some parts of the presentation of NIAQUE were confusing or unclear. How exactly is the model used for prediction if the encoder takes a variable number of inputs? At test time, do you concatenate each query point with the training dataset (or a subsample of training points) and feed them to the encoder to obtain the encoder representation? If so, have you considered various heuristics for selecting these points (e.g., nearest neighbors to the query input)?

**Questions:**

My main concern is minimal gains from multi-task transfer.
* One way to address this: is it possible to evaluate this model in the standard meta-learning setup, with entirely held-out datasets, as in [1, 2]? If these results are strong, it might better justify NIAQUE even when in the "held-in" dataset setting, NIAQUE-local and NIAQUE-global perform similarly. In the meta-learning setting, Neural Processes are the relevant baseline. If NIAQUE is competitive with them, it would encourage me to raise my score.

Other medium-priority concerns:
* In Table 1, why is Transformer-local not included? How does it perform?
* The architecture appears to resemble Neural Processes [1, 2], another neural architecture for multi-task learning / meta-learning. Could you contextualize NIAQUE with regard to them?
* The nomenclature of "co-training" seems to clash with the old-school pseudo-label style setting as in [3]. Could you instead use the established nomenclature for this setup (multi-task learning, pretraining, or meta-learning depending on the context?).

[1] Garnelo et al., 2018. Neural Processes. https://arxiv.org/abs/1807.01622

[2] Garnelo et al., 2018. Conditional Neural Processes. In ICML. https://proceedings.mlr.press/v80/garnelo18a.html

[3] Blum and Michell, 1998. Combining Labeled and Unlabeled Data with Co-Training. In COLT. https://www.cs.cmu.edu/~avrim/Papers/cotrain.pdf

---

> ### Author Response · Authors · 2024-11-24
> **Additional Experiments**
>
> Dear Reviewer p8Dt, we would like to thank you for providing insightful feedback. We have addressed your major points by (i) including the transfer learning experiment on held-out datasets and (ii) providing the Transformer-local results in Table 1. Our results demonstrate the value of pretraining NIAQUE in multi-task fashion and confirms that the learnings on various probabilistic regression tasks are generalizeable and can be transferred on unseen regression datasets using our approach. Please refer to the updated manuscript for detailed results. We are sorry that it took some time to code and execute the experiments. Please let us know if this addresses your major points. In the meanwhile, we will work on providing the response to your other points.

---

> ### Author Response · Authors · 2024-11-24
> **Response to the Neural Process question**
>
> We thank the reviewer for pointing out the Neural Process and Conditional Neural Process papers. These approaches are not directly applicable to the problem we solve, because they solve the problem in the fixed dimensional input space. For example, typical settings considered in the papers include 1d regression (curve fitting), 2d regression (image completion), classification on Omniglot. The approaches operate on the the support set of $(x,y)_i$ tuples as explained in Section of 2.4 in https://arxiv.org/pdf/1807.01622. The dimensionality of $x$ and $y$ is assumed to be fixed. The meta-learning aspect then manifests itself, for example, in the case of ominglot, when the same-dimensional 32x32 images are fed at both training and inference time, whereas the classes at train and inference time do not overlap. This is quite different from our setup, in which across different datasets the dimensionality of $x$ changes. As an example, let's take the nearest neighbor baseline used by the conditional neural process in the Omniglot experiment (Table 2 of https://proceedings.mlr.press/v80/garnelo18a/garnelo18a.pdf). Suppose we would like to implement cross-dataset knowledge transfer with variable dimensionality $x$, what is then the meaning of the nearest neighbor? None really seems to emerge naturally. Yet, our transfer learning experiment shows that positive transfer across datasets with variable-dimensional input spaces is viable. To address the reviewer's point we have included the following paragraph in related work section.
>
> Alternative approaches, such as Neural Processes \citep{garnelo2018neuralprocesses} and Conditional Neural Processes \citep{garnelo2018conditional}, also generate conditional probabilistic solutions to regression problems. However, these methods are limited to fixed-dimensional input spaces and are not directly applicable to the cross-dataset, multi-task learning problem addressed here, where datasets vary in the number of independent variables. Moreover, unlike \citet{garnelo2018neuralprocesses} and \citet{garnelo2018conditional}, our approach demonstrates the ability to transfer knowledge to entirely new datasets, even when their dependent variable domains do not overlap with the training data.

---

> ### Comment · Reviewer_p8Dt · 2024-11-25
> **Response to Author Rebuttal**
>
> Thanks for conducting the transfer learning experiments and including the Transformer-local results. Thanks also for the clarification on how this method relates to the (Conditional) Neural Process literature. I have raised my score to a 5.

---

> > ### Author Response · Authors · 2024-11-25
> > **Response to remaining points**
> >
> > Dear Reviewer p8Dt, we would like to thank you for engaging into a productive discussion and raising your score in response to our revisions! We have further revised the paper to address your remaining  points.
> >
> > First, we have worked on the terminology as you suggested and pivoted towards the multi-task formulation, replacing the co-training terminology that may not be clear.
> >
> > Second, we have clarified the operation of the architecture on vector independent variables of variable dimensionality throughout the text, but especially in the intro of Section 2.2 as follows:
> >
> > At inference time, for $i$-th observation sample, $x_i$, with variable dimensionality $N_i$ it accepts a tensor of values of dimensionality $1 \times N_i$ and a tensor of feature codes of dimensionality $1 \times N_i$, transforms, embeds and concatenates them into tensor of size $1 \times N_i \times E_{in}$. The encoder then collapses the independent variable dimension using prototype approach, resulting in output embedding of size $1 \times E$.
> >
> >
> > We would be extremely grateful to you if you could review the latest revision of the paper suggesting any further improvements to the paper you might see fit.

---

### Official Review · Reviewer_uKfu · 2024-11-04

**Soundness:** 2
**Presentation:** 3
**Contribution:** 3
**Rating:** 6
**Confidence:** 2

**Summary:**

This paper introduces NIAQUE, a new model for probabilistic regression that learns to approximate the inverse of the posterior distribution. It also introduces a new benchmark LPRM-101 with 101 datasets for multi-dataset probabilistic regression. The authors provide theoretical analysis for NIAQUE and also show that it is competitive with (or better than) prior baselines (trees, transformers) on the new benchmark. Additionally, the authors argue that NIAQUE improves over existing baselines by being more interpretable.

**Strengths:**

- To my knowledge, the benchmark is novel because it focuses on multi-dataset probabilistic regression
- NIAQUE design decisions are explained and also include some theoretical analysis
- NIAQUE is more interpretable while having competitive performance when compared with standard approaches
- Experimental results sweep over parameters and include confidence intervals

**Weaknesses:**

- While the paper introduces a new benchmark, it lacks detail or tooling that would allow others to use it. I would like to better understand how easy it would be for another researcher to use it and build off this work.
- This is a multi-dataset benchmark, but I would like to understand how the best existing single dataset approaches perform, or if the best approaches are the ones in Table 1. For example, are the top submissions to the Kaggle datasets' respective competitions significantly different from Transformers or XGBoost? How do those numbers compare with the "local" approaches in Table 1?

**Questions:**

See weaknesses.

---

> ### Author Response · Authors · 2024-11-24
> **Response**
>
> Dear Reviewer uKfu, we would like to thank you for providing insightful feedback and for positive assessment of our work. Our detailed response appears below.
>
> - The code to download benchmark datasets, pytorch dataloaders, models, training and evaluation scripts/notebooks along with trained model binaries will be publicly released if the paper is accepted. Therefore, we do not envision any problems with building on top of this work.
>
> - The focus of this paper is to demonstrate the feasibility of multi-task probabilistic learning on heterogenous regression datasets. Our most recent results in the revised paper also demonstrate positive transfer after fine-tuning on unseen datasets in this setting. To be able to demonstrate these results we used raw dataset features, uniformly across all datasets, to ensure uniformity and apple-to-apple comparison. Overwhelming majority of datasets used in our study are not Kaggle competition datasets. Kaggle competitors typically spend significant amount of time on feature engineering. Your point definitely outlines an interesting direction for future work, that of finding the automatic feature extraction methods that work synergistically with large-scale regression models helping them compete against human feature engineering approaches. This requires significant additional investigative, experiment design and theoretical work, which is outside of the scope of our current paper.

---

> > ### Comment · Reviewer_uKfu · 2024-11-26
> >
> > Thank you for your response. I choose to keep the same score.

---

### Meta-Review · Area_Chair_NQhh · 2024-12-19

**Metareview:**

This paper introduces NIAQUE, a neural network model for probabilistic regression problem that excels in any-quantile estimation across multiple datasets. It also presents a new benchmark LPRM-101, with 101 diverse regression datasets, showing that NIAQUE outperforms existing models like trees and transformers while being more interpretable and consistent in its training.

This paper got a mixed rating with borderline accepts and rejects. The authors acknowledge that NIAQUE is more interpretable with theoretical analysis and can save significant training time. The proposed LPRM-101 benchmark allows for rigorous and expansive evaluation across diverse regression tasks. Several questions and concerns are raised in the initial round, including 1. Experimental performance and comparison. Marginal gains from multi-task transfers. It is also suggested to add comparison with the best single dataset approaches. 2. Presentation and clarity issue. 3. Bias concern in co-training across datasets. After rebuttal, some reviewers raise the ratings, but it is still a borderline (e.g. after raising the rating by p8Dt, it is still a 5). Some concerns like the marginal gain from multitask transfer and the challenges of real-world application still remain. Given all these, I tend to suggest a reject given the current paper status and the ICLR standard.

**Additional Comments On Reviewer Discussion:**

A number of questions and concerns raised in the original reviews can be categorized into the following: 1) Performance and comparison, 2) Presentation and clarity, and 3) Bias and real-world application. Correspondingly, the authors added additional experiments and provided clarification for misunderstandings and unclear points. I am happy to see that the authors resolved most of the reviewers’ concerns by adding experiments and offering further explanation. However, some inherent problems, such as the marginal gain from multi-task transfers, the bias issue caused by all co-training, and the challenges of real-world application, remain not fully addressed.

---

### Decision · Program_Chairs · 2025-01-22

Reject